# Optimizing over trained GNNs via symmetry breaking

**Shiqiang Zhang**[*]
Imperial College London
London, UK

**Juan S. Campos**
Imperial College London
London, UK

**Christian Feldmann**
BASF SE
Ludwigshafen, Germany

**David Walz**
BASF SE
Ludwigshafen, Germany

**Frederik Sandfort**
BASF SE
Ludwigshafen, Germany

**Miriam Mathea**
BASF SE
Ludwigshafen, Germany

**Calvin Tsay**
Imperial College London
London, UK

**Ruth Misener**
Imperial College London
London, UK

## Abstract

Optimization over trained machine learning models has applications including: verification, minimizing neural acquisition functions, and integrating a trained surrogate into a larger decision-making problem. This paper formulates and solves optimization problems constrained by trained graph neural networks (GNNs). To circumvent the symmetry issue caused by graph isomorphism, we propose two types of symmetry-breaking constraints: one indexing a node 0 and one indexing the remaining nodes by lexicographically ordering their neighbor sets. To guarantee that adding these constraints will not remove all symmetric solutions, we construct a graph indexing algorithm and prove that the resulting graph indexing satisfies the proposed symmetry-breaking constraints. For the classical GNN architectures considered in this paper, optimizing over a GNN with a fixed graph is equivalent to optimizing over a dense neural network. Thus, we study the case where the input graph is not fixed, implying that each edge is a decision variable, and develop two mixed-integer optimization formulations. To test our symmetry-breaking strategies and optimization formulations, we consider an application in molecular design.

## 1 Introduction

Graph neural networks (GNNs) [1–3] are designed to operate on graph-structured data. By passing messages between nodes via edges (or vice versa), GNNs can efficiently capture and aggregate local information within graphs. GNN architectures including spectral approaches [4–9] and spatial approaches [10–18], are proposed based on various motivations. Due to their ability to learn non-Euclidean data structure, GNNs have recently been applied to many graph-specific tasks, demonstrating incredible performance. In drug discovery, for example, GNNs can predict molecular properties given the graph representation of molecules [16, 19–23]. Other applications using GNNs include graph clustering[24, 25], text classification [26, 27], and social recommendation [28, 29].

Although GNNs are powerful tools for these "forward" prediction tasks, few works discuss the "backward" (or inverse) problem defined on trained GNNs. Specifically, many applications motivate the inverse problem of using machine learning (ML) to predict the inputs corresponding to some output specification(s). For example, computer-aided molecular design (CAMD) [30–32] aims to

---

[*]Corresponding author: s.zhang21@imperial.ac.uk

37th Conference on Neural Information Processing Systems (NeurIPS 2023).

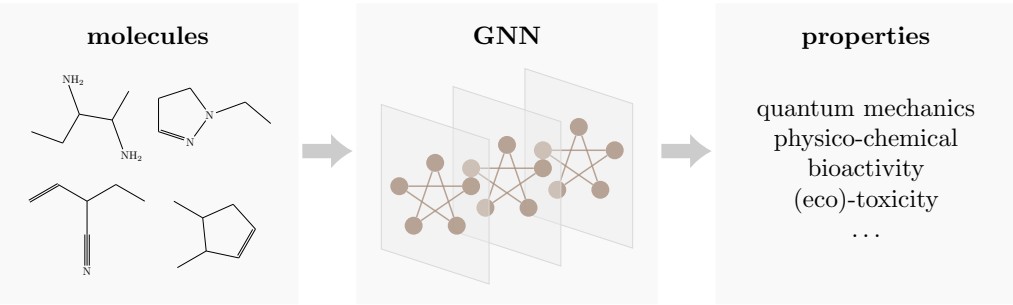

Figure 1: Illustration of forward and backward problems defined on trained GNNs.

design the optimal molecule(s) for a given target based on a trained ML model [33–36]. Several GNN-based approaches have been proposed for CAMD [37–40], however, these works still use GNNs as "forward" functions to make predictions over graph-domain inputs. Therefore, both the graph structures of inputs and the inner structures of the trained GNNs are ignored. Figure 1 conceptually depicts the difference between forward and backward/inverse problems.

Mathematical optimization over trained ML models is an area of increasing interest. In this paradigm, a trained ML model is translated into an optimization formulation, thereby enabling decision-making problems over model predictions. For continuous, differentiable models, these problems can be solved via gradient-based methods [41–44]. Mixed-integer programming (MIP) has also been proposed, mainly to support non-smooth models such as ReLU neural networks [45–50], their ensembles [51], and tree ensembles [52–54]. Optimization over trained ReLU neural networks has been especially prominent [55], finding applications such as verification [56–64], reinforcement learning [65–67], compression [68], and black-box optimization [69]. There is no reason to exclude GNNs from this rich field. Wu et al. [70] managed to verify a GNN-based job scheduler, where MIP is used to obtain tighter bounds in a forward-backward abstraction refinement framework. Mc Donald [71] built MIP formulations for GCN [6] and GraphSAGE [13] and tested their performance on a CAMD problem, which is the first work to directly apply MIP on GNNs to the best of our knowledge.

Optimization over trained GNNs involves two significant challenges. One is the symmetry issue arising from the permutation invariance of GNNs (i.e., isomorphism graphs share the same GNN output(s)). For forward problems, this invariance is preferred since it admits any graph representation for inputs. From optimization perspective, however, symmetric solutions can significantly enlarge the feasible domain and impede a branch-and-bound (B&B) search tree. One way to break symmetry in integer programs is by adding constraints to remove symmetric solutions [72–78]. The involvement of graphs not only adds complexity to the problem in terms of symmetry, but also in terms of implementation. Due to the complexity and variety of GNN architectures, a general framework is needed. This framework should be compatible with symmetry-breaking techniques.

This paper first defines optimization problems on trained GNNs. To handle the innate symmetry, we propose two sets of constraints to remove most isomorphism graphs (i.e., different indexing for any abstract graph). To guarantee that these constraints do not exclude all possible symmetries (i.e., any abstract graph should have at least one feasible indexing), we design an algorithm to index any undirected graph and prove that the resulting indexing is feasible under the proposed constraints. To encode GNNs, we build a general framework for MIP on GNNs based on the open-source Optimization and Machine Learning Toolkit (OMLT) [79]. Many classic GNN architectures are supported after proper transformations. Two formulations are provided and compared on a CAMD problem. Numerical results show the outstanding improvement of symmetry-breaking.

**Paper structure:** Section 2 introduces the problem definition, proposes the symmetry-breaking constraints, and gives theoretical guarantees. Section 3 discusses the application, implementation details, and numerical results. Section 4 concludes and discusses future work.

## 2 Methodology

### 2.1 Definition of optimization over trained GNNs

We consider optimization over a trained GNN on a given dataset $D = \{(X^i, A^i), y^i\}_{i=1}^M$. Each sample in the dataset consists of an input graph (with features $X^i$ and adjacency matrix $A^i$) and an output property $y^i$. The target of training is to approximate properties, that is:

$$GNN(X^i, A^i) \approx y^i, \ \forall 1 \le i \le M$$

When optimizing over a trained GNN, the goal is to find the input with optimal property:

$$
\begin{aligned}
(X^*, A^*) = \underset{(X,A)}{\arg\min} \ & GNN(X, A) \\
s.t. \ & f_j(X, A) \le 0, j \in \mathcal{J} \\
& g_k(X, A) = 0, k \in \mathcal{K}
\end{aligned}
\tag{OPT}
$$

where $f_j, g_k$ are problem-specific constraints and $\mathcal{J}, \mathcal{K}$ are index sets. Note that optimality of (OPT) is defined on the trained GNN instead of true properties. To simplify our presentation, we will focus on undirected graphs with node features, noting that directed graphs can be naturally transformed into undirected graphs. Likewise, using edge features does not influence our analyses. We discuss practical impacts of such modifications later.

### 2.2 Symmetry handling

For any input $(X, A)$, assume that there are $N$ nodes, giving a $N \times N$ adjacency matrix $A$ and a feature matrix $X = (X_0, X_1, \ldots, X_{N-1})^T \in \mathbb{R}^{N \times F}$, where $X_i \in \mathbb{R}^F$ contains the features for $i$-th node. For any permutation $\gamma$ over set $[N] := \{0, 1, 2 \ldots, N-1\}$, let:

$$A'_{u,v} = A_{\gamma(u),\gamma(v)}, \ X'_v = X_{\gamma(v)}, \ \forall u, v \in [N]$$

Then the permuted input $(X', A')$ has isomorphic graph structure, and therefore the same output, due to the permutation invariance of GNNs (i.e., $GNN(X, A) = GNN(X', A')$). In other words, the symmetries result from different indexing for the same graph. In general, there exist $N!$ ways to index $N$ nodes, each of which corresponds to one solution of (OPT). Mc Donald [71] proposed a set of constraints to force molecules connected, which can also be used to break symmetry in our general setting. Mathematically, they can be written as:

$$\forall v \in [N] \backslash \{0\}, \ \exists u < v, \ s.t. \ A_{u,v} = 1 \tag{S1}$$

Constraints (S1) require each node (except for node 0) to be linked with a node with smaller index. Even though constraints (S1) help break symmetry, there can still exist many symmetric solutions. To resolve, or at least relieve, this issue, we need to construct more symmetry-breaking constraints.

**Breaking symmetry at the feature level.** We can first design some constraints on the features to define a starting point for the graph (i.e., assigning index 0 to a chosen node). Note that multiple nodes may share the same features. Specifically, we define an arbitrary function $h : \mathbb{R}^F \to \mathbb{R}$, to assign a hierarchy to each node and force that node 0 has the minimal function value:

$$h(X_0) \le h(X_v), \ \forall v \in [N] \backslash \{0\} \tag{S2}$$

The choice of $h$ can be application-specific, for example, as in Section 3.1. The key is to define $h$ such that not too many nodes share the same minimal function value.

**Breaking symmetry at the graph level.** It is unlikely that constraints on features will break much of the symmetry. After adding constraints (S1) and (S2), any neighbor of node 0 could be indexed 1, then any neighbor of node 0 or 1 could be indexed 2, and so on. We want to limit the number of possible indexing. A natural idea is to account for neighbors of each node: the neighbor set of a node with smaller index should also have smaller lexicographical order.

Before further discussion, we need to define the lexicographical order. Let $\mathcal{S}(M, L)$ be the set of all non-decreasing sequences with $L$ integer elements in $[0, M]$ (i.e., $\mathcal{S}(M, L) = \{(a_1, a_2, \ldots, a_L) \mid 0 \le a_1 \le a_2 \le \cdots \le a_L \le M\}$). For any $a \in \mathcal{S}(M, L)$, denote its lexicographical order by $LO(a)$. Then for any $a, b \in \mathcal{S}(M, L)$, we have:

$$LO(a) < LO(b) \ \Leftrightarrow \ \exists l \in \{1, 2, \ldots, L\}, \ s.t. \begin{cases} a_i = b_i, & 1 \le i < l \\ a_l < b_l \end{cases} \tag{LO}$$

For any multiset $A$ with no more than $L$ integer elements in $[0, M-1]$, we first sort all elements in $A$ in a non-decreasing order, then add elements with value $M$ until the length equals to $L$. In that way, we build an injective mapping from $A$ to a sequence $a \in \mathcal{S}(M, L)$. Define the lexicographical order of $A$ by $LO(a)$. For simplicity, we use $LO(A)$ to denote the lexicographical order of $A$.

---

**Algorithm 1** Indexing algorithm

---

**Input:** $G = (V, E)$ with node set $V = \{v_0, v_1, \ldots, v_{N-1}\}$ ($N := |V|$). Denote the neighbor set of node $v$ as $\mathcal{N}(v)$, $\forall v \in V$.

$\mathcal{I}(v_0) \leftarrow 0$             $\triangleright$ Assume that $v_0$ is indexed with 0

$s \leftarrow 1$                $\triangleright$ Index for next node

$V_1^1 \leftarrow \{v_0\}$            $\triangleright$ Initialize set of indexed nodes

**while** $s < N$ **do**

  $V_2^s \leftarrow V \backslash V_1^s$            $\triangleright$ Set of unindexed nodes

  $\mathcal{N}^s(v) \leftarrow \{\mathcal{I}(u) \mid u \in \mathcal{N}(v) \cap V_1^s\}, \ \forall v \in V_2^s$    $\triangleright$ Obtain all indexed neighbors

  $rank^s(v) \leftarrow |\{LO(\mathcal{N}^s(u)) < LO(\mathcal{N}^s(v)) \mid \forall u \in V_2^s\}|, \ \forall v \in V_2^s$

               $\triangleright$ Assign a rank to each unindexed node

  $\mathcal{I}^s(v) \leftarrow \begin{cases} \mathcal{I}(v), & \forall v \in V_1^s \\ rank^s(v) + s, & \forall v \in V_2^s \end{cases}$    $\triangleright$ Assign temporary indexes

  $\mathcal{N}_t^s(v) \leftarrow \{\mathcal{I}^s(u) \mid u \in \mathcal{N}(v)\}, \ \forall v \in V_2^s$   $\triangleright$ Define temporary neighbor sets based on $\mathcal{I}^s$

  $v^s \leftarrow \arg\min_{v \in V_2^s} LO(\mathcal{N}_t^s(v))$      $\triangleright$ Neighbors of $v^s$ has minimal order

        $\triangleright$ If multiple nodes share the same minimal order, arbitrarily choose one

  $\mathcal{I}(v^s) = s$             $\triangleright$ Index $s$ to node $v^s$

  $V_1^{s+1} \leftarrow V_1^s \cup \{v^s\}$         $\triangleright$ Add $v^s$ to set of indexed nodes

  $s \leftarrow s + 1$             $\triangleright$ Next index is $s + 1$

**end while**

**Output:** $\mathcal{I}(v), v \in V$           $\triangleright$ Result indexing

---

*Remark:* The definition of $LO(\cdot)$ and Section 2.3 reuse $A$ to denote a multiset. This will not cause ambiguity since multisets are only introduced in the theoretical part, where the adjacency matrix is not involved. For brevity, we use capital letters $A, B, \ldots$ to denote multisets and lowercase letters $a, b, \ldots$ to denote the corresponding sequences in $\mathcal{S}(M, L)$.

Since we only need lexicographical orders for all neighbor sets of the nodes of the graph, let $M = N, L = N - 1$. With the definition of $LO(\cdot)$, we can represent the aforementioned idea by:
$$LO(\mathcal{N}(v)\backslash\{v+1\}) \leq LO(\mathcal{N}(v+1)\backslash\{v\}), \ \forall v \in [N-1]\backslash\{0\} \tag{S3}$$
where $\mathcal{N}(v)$ is the neighbor set of node $v$. Note that the possible edge between nodes $v$ and $v + 1$ is ignored in (S3) to include the cases that they are linked and share all neighbors with indexes less than $v$. In this case, $LO(\mathcal{N}(v+1))$ is necessarily smaller than $LO(\mathcal{N}(v))$ since $v \in \mathcal{N}(v+1)$.

Constraints (S3) exclude many ways to index the rest of the nodes, for example they reduce the possible ways to index the graph in Figure 2 from 120 to 4 ways. But, we still need to ensure that there exists at least one feasible indexing for any graph after applying constraints (S3). Algorithm 1 constructs an indexing and Section 2.3 provides a theoretical guarantee of its feasibility.

## 2.3 Theoretical guarantee

This section proves that Algorithm 1 always provides feasible indexing. We first give some properties that are used to derive an important lemma.

**Property 1.** *For any* $s = 1, 2, \ldots, N - 1$, $\mathcal{I}^s(v) \begin{cases} < s, & \forall v \in V_1^s \\ \geq s, & \forall v \in V_2^s \end{cases}$.

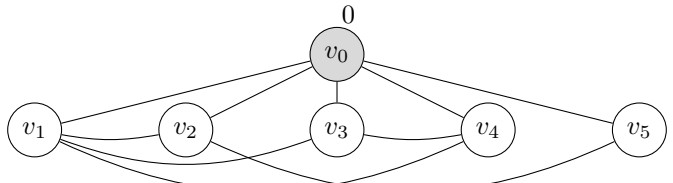

Figure 2: Consider a graph having 6 nodes with different features. Without any constraints, there are $6! = 720$ ways to index it. Utilizing (S1) to force connectivity results in 636 ways. Using (S2) to choose $v_0$ to be indexed 0, then there are $5! = 120$ ways. Finally, applying (S3) to order the rest of the nodes, there are only 4 ways, 2 of which can be derived from Algorithm 1. For details on using Algorithm 1 to index this graph, see Appendix A.2.

**Property 2.** *For any $s_1, s_2 = 1, 2, \ldots, N-1$,*

$$s_1 \leq s_2 \Rightarrow \mathcal{N}^{s_1}(v) = \mathcal{N}^{s_2}(v) \cap [s_1], \ \forall v \in V_2^{s_2}$$

**Property 3.** *Given any multisets $A, B$ with no more than $L$ integer elements in $[0, M-1]$, we have:*

$$LO(A) \leq LO(B) \Rightarrow LO(A \cap [m]) \leq LO(B \cap [m]), \ \forall m = 1, 2, \ldots, M$$

Using these properties, we can prove Lemma 1, which shows that if the final index of node $u$ is smaller than $v$, then at each iteration, the temporary index assigned to $u$ is not greater than $v$.

**Lemma 1.** *For any two nodes $u$ and $v$,*

$$\mathcal{I}(u) < \mathcal{I}(v) \Rightarrow \mathcal{I}^s(u) \leq \mathcal{I}^s(v), \ \forall s = 1, 2, \ldots, N-1$$

Now we can prove Theorem 1.

**Theorem 1.** *Given any undirected graph $G = (V, E)$ with one node indexed $0$. The indexing yielded from Algorithm 1 satisfies (S3).*

*Proof.* Assume that constraints (S3) are not satisfied and the minimal index that violates (S3) is $s$. Denote nodes with index $s$ and $s+1$ by $u, v$ respectively (i.e., $\mathcal{I}(u) = s, \mathcal{I}(v) = s+1$).

Let $\mathcal{N}(u) \backslash \{v\} := \{u_1, u_2, \ldots, u_m\}$ be all neighbors of $u$ except for $v$, where:

$$\mathcal{I}(u_i) < \mathcal{I}(u_{i+1}), \ \forall i = 1, 2, \ldots, m-1$$

Similarly, let $\mathcal{N}(v) \backslash \{u\} := \{v_1, v_2, \ldots, v_n\}$ be all neighbors of $v$ except for $u$, where:

$$\mathcal{I}(v_j) < \mathcal{I}(v_{j+1}), \ \forall j = 1, 2, \ldots, n-1$$

Denote the sequences in $\mathcal{S}(N, N-1)$ corresponding to sets $\{\mathcal{I}(u_i) \mid 1 \leq i \leq m\}$ and $\{\mathcal{I}(v_j) \mid 1 \leq j \leq n\}$ by $a = (a_1, a_2, \ldots, a_{N-1}), b = (b_1, b_2, \ldots, b_{N-1})$. By definition of $LO(\cdot)$:

$$a_i = \begin{cases} \mathcal{I}(u_i), & 1 \leq i \leq m \\ N, & m < i < N \end{cases}, \ b_j = \begin{cases} \mathcal{I}(v_j), & 1 \leq j \leq n \\ N, & n < j < N \end{cases}$$

Since nodes $u$ and $v$ violate constraints (S3), there exists a position $1 \leq k \leq N-1$ satisfying:

$$\begin{cases} a_i = b_i, & \forall 1 \leq i < k \\ a_k > b_k \end{cases}$$

from where we know that nodes $u$ and $v$ share the first $k-1$ neighbors (i.e. $u_i = v_i, \ \forall 1 \leq i < k$). From $b_k < a_k \leq N$ we know that node $v$ definitely has its $k$-th neighbor node $v_k$. Also, note that $v_k$ is not a neighbor of node $u$. Otherwise, we have $u_k = v_k$ and then $a_k = \mathcal{I}(u_k) = \mathcal{I}(v_k) = b_k$.

*Case 1:* If $a_k = N$, that is, node $u$ has $k-1$ neighbors.

In this case, node $v$ has all neighbors of node $u$ as well as node $v_k$. Therefore, we have:

$$LO(\mathcal{N}_t^s(u)) > LO(\mathcal{N}_t^s(v))$$

which violates the fact that node $u$ is chosen to be indexed $s$ at $s$-th iteration of Algorithm 1.

*Case 2:* If $a_k < N$, that is, node $u$ has nodes $u_k$ as its $k$-th neighbor.

Since $\mathcal{I}(u_k) = a_k > b_k = \mathcal{I}(v_k)$, we can apply Lemma 1 on node $u_k$ and $v_k$ at $(s+1)$-th iteration, to obtain

$$\mathcal{I}^{s+1}(u_k) \geq \mathcal{I}^{s+1}(v_k) \tag{$\geq$}$$

Similarly, if we apply Lemma 1 to all the neighbors of node $u$ and node $v$ at $s$-th iteration, we have:

$$\mathcal{I}^s(u_i) \leq \mathcal{I}^s(u_{i+1}), \ \forall i = 1, 2, \ldots, m-1$$
$$\mathcal{I}^s(v_j) \leq \mathcal{I}^s(v_{j+1}), \ \forall j = 1, 2, \ldots, n-1$$

Given that $a_k = \mathcal{I}(u_k)$ is the $k$-th smallest number in $a$, we conclude that $\mathcal{I}^s(u_k)$ is equal to the $k$-th smallest number in $\mathcal{N}_t^s(u)$. Likewise, $\mathcal{I}^s(v_k)$ equals to the $k$-th smallest number in $\mathcal{N}_t^s(v)$. Meanwhile, $\mathcal{I}^s(u_i) = \mathcal{I}^s(v_i)$ since $u_i = v_i$, $\forall 1 \leq i < k$. After comparing the lexicographical orders between of $\mathcal{N}_t^s(u)$ and $\mathcal{N}_t^s(v)$ (with the same $k-1$ smallest elements, $\mathcal{I}^s(u_k)$ and $\mathcal{I}^s(v_k)$ as the $k$-th smallest element, respectively), node $u$ is chosen. Therefore, we have:

$$\mathcal{I}^s(u_k) \leq \mathcal{I}^s(v_k)$$

from which we know that:

$$LO(\mathcal{N}^s(u_k)) \leq LO(\mathcal{N}^s(v_k))$$

At $(s+1)$-th iteration, node $u_k$ has one more indexed neighbor (i.e., node $u$ with index $s$), while node $v_k$ has no new indexed neighbor. Thus we have:

$$LO(\mathcal{N}^{s+1}(u_k)) = LO(\mathcal{N}^s(u_k) \cup \{s\}) < LO(\mathcal{N}^s(u_k)) \leq LO(\mathcal{N}^s(v_k)) = LO(\mathcal{N}^{s+1}(v_k))$$

which yields:

$$\mathcal{I}^{s+1}(u_k) < \mathcal{I}^{s+1}(v_k) \tag{$<$}$$

The contradiction between $(\geq)$ and $(<)$ completes this proof. $\square$

Given any undirected graph with node features, after using (S2) to choose node 0, Theorem 1 guarantees that there exists at least one indexing satisfying (S3). However, when applying (S1) to force a connected graph, we need to show the compatibility between (S1) and (S3), as shown in Lemma 2. Appendix A provides proofs of properties and lemmas.

**Lemma 2.** *For any undirected, connected graph $G = (V, E)$, if one indexing of $G$ sastifies* (S3), *then it satisfies* (S1).

Note that (S1) - (S3) are not limited to optimizing over trained GNNs. In fact, they can be employed in generic graph-based search problems, as long as there exists a symmetry issue caused by graph isomorphism. GNNs can be generalized to all permutation invariant functions defined over graphs. Although the next sections apply MIP formulations to demonstrate the symmetry-breaking constraints, we could alternatively use a constraint programming [80] or satisfiability [81] paradigm. For example, we could have encoded the Elgabou and Frisch [82] constraints in Reluplex [83].

## 2.4 Connection & Difference to the symmetry-breaking literature

Typically, MIP solvers detect symmetry using graph automorphism, for example SCIP [84] uses BLISS [85, 86], and both Gurobi [87] and SCIP break symmetry using orbital fixing [88] and orbital pruning [89]. When a MIP solver detects symmetry, the only graph available is the graph formed by the variables and constraints in the MIP.

Our symmetries come from alternative indexing of abstract nodes. Each indexing results in an isomorphic graph. In MIP, however, each indexing corresponds to an element of a much larger symmetry group defined on all variables, including node features ($O(NF)$), adjacency matrix ($O(N^2)$), model (e.g., a GNN), and problem-specific features. For instance, in the first row of Table 6, the input graph has $N = 4$ nodes, but the MIP has $616 + 411$ variables. We only need to consider a permutation group with $N! = 24$ elements. However, because the MIP solver does not have access to the input graph structure, it needs to consider all possible automorphic graphs with $616 + 411$ nodes. By adding constraints (S1) - (S3), there is no need to consider the symmetry group of all variables to find a much smaller subgroup corresponding to the permutation group defined on abstract nodes.

The closest setting to ours is distributing $m$ different jobs to $n$ identical machines and then minimizing the total cost. Binary variable $A_{i,j}$ denotes if job $i$ is assigned to machine $j$. The requirement is that each job can only be assigned to one machine (but each machine can be assigned to multiple jobs). Symmetries come from all permutations of machines. This setting appears in noise dosage problems [90, 91], packing and partitioning orbitopes [92, 93], and scheduling problems [94]. However, requiring that the sum of each row in $A_{i,j}$ equals to 1 simplifies the problem. By forcing decreasing lexicographical orders for all columns, the symmetry issue is handled well. Constraints (S3) can be regarded as a non-trivial generalization of these constraints from a bipartite graph to an arbitrary undirected graph: following Algorithm 1 will produce the same indexing documented in [90–94].

## 2.5 Mixed-integer formulations for optimizing over trained GNNs

As mentioned before, there are many variants of GNNs [4–18] with different theoretical and practical motivations. Although it is possible to build a MIP for a specific GNN from the scratch (e.g., [71]), a general definition that supports multiple architectures is preferable.

### 2.5.1 Definition of GNNs

We define a GNN with $L$ layers as follows:

$$GNN : \underbrace{\mathbb{R}^{d_0} \otimes \cdots \otimes \mathbb{R}^{d_0}}_{|V| \text{ times}} \to \underbrace{\mathbb{R}^{d_L} \otimes \cdots \otimes \mathbb{R}^{d_L}}_{|V| \text{ times}}$$

where $V$ is the set of nodes of the input graph.

Let $\boldsymbol{x}_v^{(0)} \in \mathbb{R}^{d_0}$ be the input features for node $v$. Then, the $l$-th layer ($l = 1, 2, \ldots, L$) is defined by:

$$\boldsymbol{x}_v^{(l)} = \sigma \left( \sum_{u \in \mathcal{N}(v) \cup \{v\}} \boldsymbol{w}_{u \to v}^{(l)} \boldsymbol{x}_u^{(l-1)} + \boldsymbol{b}_v^{(l)} \right), \ \forall v \in V \tag{$\star$}$$

where $\mathcal{N}(v)$ is the set of all neighbors of $v$, $\sigma$ could be identity or any activation function.

With linear aggregate functions such as sum and mean, many classic GNN architectures could be rewritten in form ($\star$), for example: Spectral Network [4], ChebNet [5], GCN [6], Neural FPs [10], DCNN [11], PATCHY-SAN [12], GraphSAGE [13], and MPNN [16].

### 2.5.2 Mixed-integer optimization formulations for non-fixed graphs

If the graph structure for inputs is given and fixed, then ($\star$) is equivalent to a fully connected layer, whose MIP formulations are already well-established [46, 49]. But if the graph structure is non-fixed (i.e., the elements in adjacency matrix are also decision variables), two issues arise in ($\star$): (1) $\mathcal{N}(v)$ is not well-defined; (2) $\boldsymbol{w}_{u \to v}^{(l)}$ and $\boldsymbol{b}_v^{(l)}$ may be not fixed and contain the graph's information. Assuming that the weights and biases are constant, we can build two MIP formulations to handle the first issue.

The first formulation comes from observing that the existence of edge from node $u$ to node $v$ determines the contribution link from $\boldsymbol{x}_u^{(l-1)}$ to $\boldsymbol{x}_v^{(l)}$. Adding binary variables $e_{u \to v}$ for all $u, v \in V$, we can then formulate the GNNs with bilinear constraints:

$$\boldsymbol{x}_v^{(l)} = \sigma \left( \sum_{u \in V} e_{u \to v} \boldsymbol{w}_{u \to v}^{(l)} \boldsymbol{x}_u^{(l-1)} + \boldsymbol{b}_v^{(l)} \right), \ \forall v \in V \tag{bilinear}$$

Introducing this nonlinearity results in a mixed-integer quadratically constrained optimization problem (MIQCP), which can be handled by state-of-the-art solvers such as Gurobi [87].

The second formulation generalizes the big-M formulation for GraphSAGE in [71] to all GNN architectures satisfying ($\star$) and the assumption of constant weights and biases. Instead of using binary variables to directly control the existence of contributions between nodes, auxiliary variables $\boldsymbol{z}_{u \to v}^{(l-1)}$ are introduced to represent the contribution from node $u$ to node $v$ in the $l$-th layer:

$$\boldsymbol{x}_v^{(l)} = \sigma \left( \sum_{u \in V} \boldsymbol{w}_{u \to v}^{(l)} \boldsymbol{z}_{u \to v}^{(l-1)} + \boldsymbol{b}_v^{(l)} \right), \ \forall v \in V \tag{big-M}$$

where:

$$\boldsymbol{z}_{u \to v}^{(l-1)} = \begin{cases} 0, & e_{u \to v} = 0 \\ \boldsymbol{x}_u^{(l-1)}, & e_{u \to v} = 1 \end{cases}$$

Assuming that each feature is bounded, the definition of $\boldsymbol{z}_{u \to v}^{(l-1)}$ can be reformulated using big-M:

$$\boldsymbol{x}_u^{(l-1)} - \boldsymbol{M}_u^{(l-1)}(1 - e_{u \to v}) \le \boldsymbol{z}_{u \to v}^{(l-1)} \le \boldsymbol{x}_u^{(l-1)} + \boldsymbol{M}_u^{(l-1)}(1 - e_{u \to v})$$
$$-\boldsymbol{M}_u^{(l-1)} e_{u \to v} \le \boldsymbol{z}_{u \to v}^{(l-1)} \le \boldsymbol{M}_u^{(l-1)} e_{u \to v}$$

where $|\boldsymbol{x}_u^{(l-1)}| \le \boldsymbol{M}_u^{(l-1)}, e_{u \to v} \in \{0, 1\}$. By adding extra continuous variables and constraints, as well as utilizing the bounds for all features, the big-M formulation replaces the bilinear constraints with linear constraints. Section 3 numerically compares these two formulations.

## 3 Computational experiments

We performed all experiments on a 3.2 GHz Intel Core i7-8700 CPU with 16 GB memory. GNNs are implemented and trained in *PyG* [95]. MIP formulations for GNNs and CAMD are implemented based on OMLT [79], and are optimized by Gurobi 10.0.1 [87] with default relative MIP optimality gap (i.e., $10^{-4}$). The code is available at https://github.com/cog-imperial/GNN_MIP_CAMD. These are all engineering choices and we could have, for example, extended software other than OMLT to translate the trained GNNs into an optimization solver [96, 97].

### 3.1 Mixed-integer optimization formulation for molecular design

MIP formulations are well-established in the CAMD literature [98–103]. The basic idea is representing a molecule as a graph, creating variables for each atom (or group of atoms), using constraints to preserve graph structure and satisfy chemical requirements. A score function is usually given as the optimization target. Instead of using knowledge from experts or statistics to build the score function, GNNs (or other ML models) could be trained from datasets to replace score functions. Here we provide a general and flexible framework following the MIP formulation for CAMD in [71]. Moreover, by adding meaningful bounds and breaking symmetry, our formulation could generate more reasonable molecules with less redundancy. Due to space limitation, we briefly introduce the formulation here (see Appendix B for the full MIP formulation).

To design a molecule with $N$ atoms, we define $N \times F$ binary variables $X_{v,f}, v \in [N], f \in [F]$ to represent $F$ features for $N$ atoms. Hydrogen atoms are not counted in since they can implicitly considered as node features. Features include types of atom, number of neighbors, number of hydrogen atoms associated, and types of adjacent bonds. For graph structure of molecules, we add three sets of binary variables $A_{u,v}, DB_{u,v}, TB_{u,v}$ to denote the type of bond (i.e., any bond, double bond, and triple bond) between atom $u$ and atom $v$, where $A$ is the adjacency matrix.

To design reasonable molecules, constraints (C1) - (C21) handle structural feasibility following [71]. Additionally, we propose new constraints to bound the number of each type of atoms (C22), double bonds (C23), triple bonds (C24), and rings (C25). In our experiments, we calculate these bounds based on the each dataset itself so that each molecule in the dataset will satisfy these bounds (see Appendix C.1 for details). By setting proper bounds, we can control the composition of the molecule, and avoid extreme cases such as all atoms being set to oxygen, or a molecule with too many rings or double/triple bounds. In short, constraints (C22) - (C25) provide space for chemical expertise and practical requirements. Furthermore, our formulation could be easily applied to datasets with different types of atoms by only changing the parameters in Appendix B.1. Moreover, group representation of molecules [98, 102] is also compatible with this framework. The advantage is that all constraints can be reused without any modification.

Among constraints (C1) - (C25) (as shown in Appendix B.3), constraints (C5) are the realization of (S1). Except for (C5), these structural constraints are independent of the graph indexing. Therefore, we can compatibly implement constraints (S2) and (S3) to break symmetry.

Corresponding to (S2), we add the following constraints over features:

$$\sum_{f \in [F]} 2^{F-f-1} \cdot X_{0,f} \le \sum_{f \in [F]} 2^{F-f-1} \cdot X_{v,f}, \ \forall v \in [N] \backslash \{0\} \tag{C26}$$

where $2^{F-f-1}, f \in [F]$ are coefficients to help build a bijective $h$ between all possible features and all integers in $[0, 2^F - 1]$. These coefficients are also called "universal ordering vector" in [72, 78].

On graph level, constraints (S3) can be equivalently rewritten as:

$$\sum_{u \neq v, v+1} 2^{N-u-1} \cdot A_{u,v} \geq \sum_{u \neq v, v+1} 2^{N-u-1} \cdot A_{u,v+1}, \ \forall v \in [N-1]\backslash\{0\} \tag{C27}$$

Similarly, the coefficients $2^{N-u-1}, u \in [N]$ are used to build a bijective mapping between all possible sets of neighbors and all integers in $[0, 2^N - 1]$.

For illustration purposes, we can view CAMD as two separate challenges, where the first one uses structural constraints to design reasonable molecules (including all symmetric solutions), and the second one uses symmetry-breaking constraints to remove symmetric solutions. Note that the diversity of solutions will not change after breaking symmetry, because each molecule corresponds to at least one solution (guaranteed by Section 2.3).

## 3.2 Counting feasible structures: performance of symmetry-breaking

We choose two datasets QM7 [104, 105] and QM9 [106, 107] from CAMD literature to test the proposed methods. See Appendix C.1 for more information about QM7 and QM9. To test the efficiency of our symmetry-breaking techniques, we build a MIP formulation for CAMD and count all feasible structures for different $N$. By setting PoolSearchMode=2, PoolSolutions=$10^9$, Gurobi can find many (up to $10^9$) feasible solutions to fill in the solution pool.

Table 1 shows the performance of our symmetry-breaking constraints (S2) and (S3) comparing to the baseline of (S1). Without adding (S1), we need to count the number of any graph with compatible features. Even ignoring features, the baseline would be $2^{\frac{N(N-1)}{2}}$. This number will be much larger after introducing features, which loses the meaning as a baseline, so we use (S1) as the baseline.

Table 1: Numbers of feasible solutions. The time limit is $48$ hours. At least $2.5 \times 10^6$ solutions are found for each time out (t.o.). For each dataset, the last column reports the percentage of removed symmetric solutions after adding (S2) and (S3) to the baseline of (S1). Higher percentage means breaking more symmetries.

| $N$ | QM7 | | | | QM9 | | | |
| | (S1) | (S1) - (S2) | (S1) - (S3) | (%) | (S1) | (S1) - (S2) | (S1) - (S3) | (%) |
|---|---|---|---|---|---|---|---|---|
| 2 | 17 | 10 | 10 | 41 | 15 | 9 | 9 | 40 |
| 3 | 112 | 37 | 37 | 67 | 175 | 54 | 54 | 69 |
| 4 | $3,323$ | 726 | 416 | 87 | $4,536$ | $1,077$ | 631 | 86 |
| 5 | $67,020$ | $11,747$ | $3,003$ | 96 | $117,188$ | $21,441$ | $5,860$ | 95 |
| 6 | t.o. | $443,757$ | $50,951$ | $\geq 98$ | t.o. | $527,816$ | $59,492$ | $\geq 98$ |
| 7 | t.o. | t.o. | $504,952$ | $*$ | t.o. | t.o. | $776,567$ | $*$ |
| 8 | t.o. | t.o. | t.o. | $*$ | t.o. | t.o. | t.o. | $*$ |

## 3.3 Optimizing over trained GNNs for molecular design

For each dataset, we train a GNN with two GraphSAGE layers followed by an add pooling layer, and three dense layers. Details about their structures and training process are shown in Appendix C.2. For statistical consideration, we train 10 models with different random seeds and use the 5 models with smaller losses for optimization. Given $N$ ($\in \{4, 5, 6, 7, 8\}$), and a formulation ($\in \{$bilinear, bilinear+BrS, big-M, big-M+BrS$\}$), where "+BrS" means adding symmetry-breaking constraints (C26) and (C27), we solve the corresponding optimization problem 10 times with different random seeds in Gurobi. This means that there are 50 runs for each $N$ and formulation.

Figure 3 shows the significant improvement of symmetry-breaking. Considering the average solving time, big-M performs better. For the bilinear constraints, Gurobi 10.0.1 appears to transform bilinear constraints into linear constraints. Appendix C.3 shows that, after Gurobi's presolve stage, the big-M formulation has more continuous variables but fewer binary variables compared to the bilinear formulation. The fewer binary variables after presolve may explain the better big-M performance.

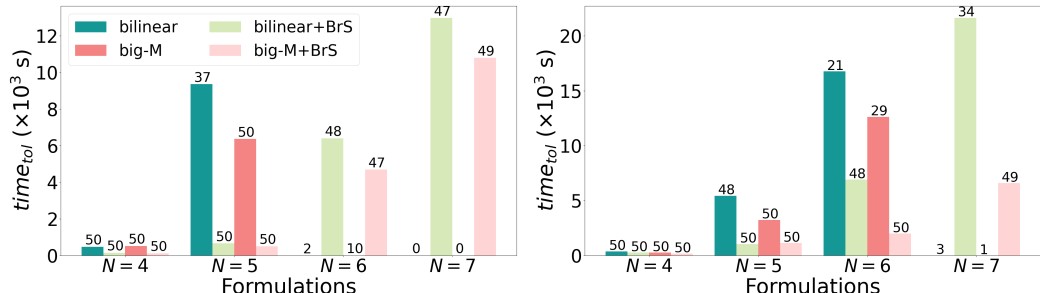

Figure 3: These graphs report the time $time_{tol}$ to achieve relative MIP optimality gap $10^{-4}$ averaged over the number of successful runs among 50 runs. We plot a bar for each formulation at different values of $N$. The left- and right-hand graphs correspond to datasets QM7 and QM9, respectively. The number of runs that achieve optimality is shown above each bar. Results with fewer than 20 successful runs do not have a bar because so few runs terminated. The time limit is 10 hours.

In addition to the average solving time, we compare the performance of two formulations with breaking symmetry in each run. In 341 runs out of 500 runs (i.e., 50 runs for each dataset $\in$ {QM7, QM9} with each $N \in \{4, 5, 6, 7, 8\}$), the big-M formulation achieves optimality faster than the bilinear formulation. Additionally, Gurobi uses much of the solving time to improve the bounds. Table 6 and 7 in Appendix C.3 report $time_{opt}$ that denotes the first time to find the optimal solution. In 292 runs, the big-M formulation finds the optimal solution earlier than the bilinear formulation.

## 4 Discussion & Conclusion

We introduce optimization over trained GNNs and propose constraints to break symmetry. We prove that there exists at least one indexing (resulting from Algorithm 1) satisfying these constraints. Numerical results show the significant improvement after breaking symmetry. These constraints are not limited to the problem (i.e., optimizing trained GNNs), technique (i.e., MIP), and application (i.e., CAMD) used in this work. For example, one can incorporate them into genetic algorithms instead of MIP, or replace the GNN by artificially-designed score functions or other ML models. In other graph-based decision-making problems, as long as the symmetry issue caused by graph isomorphism exists, these constraints could be used to break symmetry. Moreover, the proposed frameworks for building MIP for GNNs as well as CAMD provide generality and flexibility for more problems.

**Limitations.** Assuming constant weights and biases excludes some GNN types. This limitation is an artifact of our framework and it is possible to build MIP formulations for some other architectures. Using MIP may limit GNN size, for instance, edge features may enlarge the optimization problem.

**Future work.** One direction is to make the MIP-GNN framework more general, such as adding edge features, supporting more GNN architectures, and developing more efficient formulations. Another direction is using the symmetry-breaking constraints in graph-based applications beyond trained GNNs because we can consider any function that is invariant to graph isomorphism. To facilitate further CAMD applications, more features such as aromacity and formal charge could be incorporated. Also, optimizing lead structures in a defined chemical space is potential to design large molecules.

## Acknowledgements

This work was supported by the Engineering and Physical Sciences Research Council [grant numbers EP/W003317/1 and EP/T001577/1], an Imperial College Hans Rausing PhD Scholarship to SZ, a BASF/RAEng Research Chair in Data-Driven Optimisation to RM, and an Imperial College Research Fellowship to CT.

We really appreciate the comments and suggestions from reviewers and meta-reviewers during the peer review process, which were very helpful to clarify and improve this paper. We also thank Guoliang Wang for the discussion about graph indexing.

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

# A Theoretical guarantee: supplementary

## A.1 Proofs of properties and lemma

**Property 1.** *For any* $s = 1, 2, \ldots, N-1$, $\mathcal{I}^s(v) \begin{cases} < s, & \forall v \in V_1^s \\ \geq s, & \forall v \in V_2^s \end{cases}$.

*Proof.* At the $s$-th iteration, nodes in $V_1^s$ have been indexed by $0, 1, \ldots, s-1$. Therefore, if $v \in V_1^s$, then $\mathcal{I}^s(v) = \mathcal{I}(v) < s$. If $v \in V_2^s$, since $\mathcal{I}^s(v)$ is the sum of $s$ and $rank^s(v)$ (which is non-negative), then we have $\mathcal{I}^s(v) \geq s$. $\square$

**Property 2.** *For any* $s_1, s_2 = 1, 2, \ldots, N-1$,

$$s_1 \leq s_2 \Rightarrow \mathcal{N}^{s_1}(v) = \mathcal{N}^{s_2}(v) \cap [s_1], \ \forall v \in V_2^{s_2}$$

*Proof.* First, $\mathcal{N}^{s_1}(\cdot)$ is well-defined on $V_2^{s_2}$ since $V_2^{s_2} \subset V_2^{s_1}$. By definitions of $\mathcal{N}^s(\cdot)$ and $V_1^s$, for any $v \in V_2^{s_2}$ we have:

$$\begin{aligned}
\mathcal{N}^{s_1}(v) &= \{\mathcal{I}(u) \mid u \in \mathcal{N}(v) \cap V_1^{s_1}\} \\
&= \{\mathcal{I}(u) \mid u \in \mathcal{N}(v) \cap V_1^{s_1} \cap V_1^{s_2}\} \\
&= \{\mathcal{I}(u) \mid u \in (\mathcal{N}(v) \cap V_1^{s_2}) \cap V_1^{s_1}\} \\
&= \{\mathcal{I}(u) \mid u \in \mathcal{N}(v) \cap V_1^{s_2}, \mathcal{I}(u) < s_1\} \\
&= \{\mathcal{I}(u) \mid u \in \mathcal{N}(v) \cap V_1^{s_2}\} \cap [s_1] \\
&= \mathcal{N}^{s_2}(v) \cap [s_1]
\end{aligned}$$

where the second equation uses the fact that $V_1^{s_1} \subset V_1^{s_2}$, and the fourth equation holds since $u \in V_1^{s_1} \Leftrightarrow \mathcal{I}(u) < s_1$ (using Property 1). $\square$

**Property 3.** *Given any multisets* $A, B$ *with no more than* $L$ *integer elements in* $[0, M-1]$, *we have:*

$$LO(A) \leq LO(B) \Rightarrow LO(A \cap [m]) \leq LO(B \cap [m]), \ \forall m = 1, 2, \ldots, M$$

*Proof.* By the definition of lexicographical order for multisets, denote the corresponding sequence to $A, B, A \cap [m], B \cap [m]$ by $a, b, a^m, b^m \in \mathcal{S}(M, L)$, respectively. Then it is equivalent to show that:

$$LO(a) \leq LO(b) \Rightarrow LO(a^m) \leq LO(b^m), \ \forall m = 1, 2, \ldots, M$$

Let $a = (a_1, a_2, \ldots, a_L)$ and $b = (b_1, b_2, \ldots, b_L)$. If $LO(a) = LO(b)$, then $a = b$ and $a^m = b^m$. Thus $LO(a^m) = LO(b^m)$. Otherwise, if $LO(a) < LO(b)$, then there exists $1 \leq l \leq L$ such that:

$$\begin{cases} a_i = b_i, & \forall 1 \leq i < l \\ a_l < b_l \end{cases}$$

If $m \leq a_l$, then $a^m = b^m$, which means $LO(a^m) = LO(b^m)$. Otherwise, if $m > a_l$, then $a^m$ and $b^m$ share the same first $l-1$ elements. But the $l$-th element of $a^m$ is $a_l$, while the $l$-th element of $b_m$ is either $b_l$ or $M$. In both cases, we have $LO(a^m) < LO(b^m)$. $\square$

**Lemma 1.** *For any two nodes* $u$ *and* $v$,

$$\mathcal{I}(u) < \mathcal{I}(v) \Rightarrow \mathcal{I}^s(u) \leq \mathcal{I}^s(v), \ \forall s = 1, 2, \ldots, N-1$$

*Proof. Case 1:* If $\mathcal{I}(u) < \mathcal{I}(v) < s$, then:

$$\mathcal{I}^s(u) = \mathcal{I}(u) < \mathcal{I}(v) = \mathcal{I}^s(v)$$

*Case 2:* If $\mathcal{I}(u) < s \leq \mathcal{I}(v)$, then $\mathcal{I}^s(u) = \mathcal{I}(u)$ and:

$$\mathcal{I}^s(v) \geq s > \mathcal{I}(u) = \mathcal{I}^s(u)$$

where Property 1 is used.

*Case 3:* If $s \leq \mathcal{I}(u) < \mathcal{I}(v)$, at $\mathcal{I}(u)$-th iteration, $u$ is chosen to be indexed $\mathcal{I}(u)$. Thus:

$$LO(\mathcal{N}_t^{\mathcal{I}(u)}(u)) \leq LO(\mathcal{N}_t^{\mathcal{I}(u)}(v))$$

According to the definition of $\mathcal{N}^{\mathcal{I}(u)}(\cdot)$ and $\mathcal{N}_t^{\mathcal{I}(u)}(\cdot)$, we have:

$$\mathcal{N}^{\mathcal{I}(u)}(u) = \mathcal{N}_t^{\mathcal{I}(u)}(u) \cap [\mathcal{I}(u)], \, \mathcal{N}^{\mathcal{I}(u)}(v) = \mathcal{N}_t^{\mathcal{I}(u)}(v) \cap [\mathcal{I}(u)]$$

Using Property 3 (with $A = \mathcal{N}_t^{\mathcal{I}(u)}(u)$, $B = \mathcal{N}_t^{\mathcal{I}(u)}(v)$, $m = \mathcal{I}(u)$) yields:

$$LO(\mathcal{N}^{\mathcal{I}(u)}(u)) \leq LO(\mathcal{N}^{\mathcal{I}(u)}(v))$$

Apply Property 2 (with $s_1 = s, s_2 = \mathcal{I}(u)$) for $u$ and $v$, we have:

$$\mathcal{N}^s(u) = \mathcal{N}^{\mathcal{I}(u)}(u) \cap [s], \, \mathcal{N}^s(v) = \mathcal{N}^{\mathcal{I}(u)}(v) \cap [s]$$

Using Property 3 again (with $A = N^{\mathcal{I}(u)}(u)$, $B = N^{\mathcal{I}(u)}(v)$, $m = s$) gives:

$$LO(\mathcal{N}^s(u)) \leq LO(\mathcal{N}^s(v))$$

Recall the definition of $\mathcal{I}^s(\cdot)$, we have:

$$\mathcal{I}^s(u) \leq \mathcal{I}^s(v)$$

$\square$

**Lemma 2.** *For any undirected, connected graph $G = (V, E)$, if one indexing of $G$ sastifies* (S3), *then it satisfies* (S1).

*Proof.* Node 0 itself is a connected graph. Assume that the subgraph induced by nodes $\{0, 1, \ldots, v\}$ is connected, it suffices to show that the subgraph induced by nodes $\{0, 1, \ldots, v+1\}$ is connected. Equivalently, we need to prove that there exists $u < v + 1$ such that $A_{u,v+1} = 1$.

Assume that $A_{u,v+1} = 0, \, \forall u < v + 1$. Since $G$ is connected, there exists $v' > v + 1$ such that:

$$\exists u < v + 1, \, s.t. \, A_{u,v'} = 1$$

Then we know:

$$\mathcal{N}(v+1) \cap [v+1] = \emptyset, \, \mathcal{N}(v') \cap [v+1] \neq \emptyset$$

Recall the definition of $LO(\cdot)$, we obtain that:

$$LO(\mathcal{N}(v+1) \cap [v+1]) > LO(\mathcal{N}(v') \cap [v+1]) \tag{>}$$

Since the indexing satisfies (S3), then we have:

$$LO(\mathcal{N}(w) \backslash \{w+1\}) \leq LO(\mathcal{N}(w+1) \backslash \{w\}), \, \forall v < w < v'$$

Applying Property 3:

$$LO((\mathcal{N}(w) \backslash \{w+1\}) \cap [v+1]) \leq LO((\mathcal{N}(w+1) \backslash \{w\}) \cap [v+1]), \, \forall v < w < v'$$

Note that $w > v$. Therefore,

$$LO(\mathcal{N}(w) \cap [v+1]) \leq LO(\mathcal{N}(w+1) \cap [v+1]), \, \forall v < w < v'$$

Choosing $w = v + 1$ gives:

$$LO(\mathcal{N}(v+1) \cap [v+1]) \leq LO(\mathcal{N}(v') \cap [v+1]) \tag{$\leq$}$$

The contradiction between (>) and ($\leq$) completes the proof. $\square$

## A.2 Example for applying Algorithm 1

Given a graph with $N = 6$ nodes as shown in Figure 4, where $v_0$ is already indexed 0. We provide details of Algorithm 1 by indexing the rest 5 nodes step by step for further illustration.

Before indexing, we first calculate the neighbor sets for each node:

$$\mathcal{N}(v_0) = \{v_1, v_2, v_3, v_4, v_5\}, \mathcal{N}(v_1) = \{v_0, v_2, v_3, v_4\}, \mathcal{N}(v_2) = \{v_0, v_1, v_5\}$$
$$\mathcal{N}(v_3) = \{v_0, v_1, v_4\}, \mathcal{N}(v_4) = \{v_0, v_1, v_3\}, \mathcal{N}(v_5) = \{v_0, v_2\}$$

$s = 1 : V_1^1 = \{v_0\}, V_2^1 = \{v_1, v_2, v_3, v_4, v_5\}$

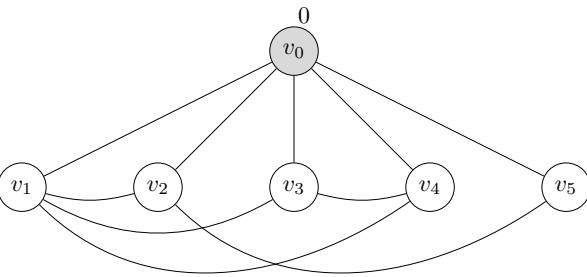

Figure 4: $V_1^1 = \{v_0\}$

Obtain indexed neighbors to each unindexed node:

$$\mathcal{N}^1(v_1) = \{0\}, \mathcal{N}^1(v_2) = \{0\}, \mathcal{N}^1(v_3) = \{0\}, \mathcal{N}^1(v_4) = \{0\}, \mathcal{N}^1(v_5) = \{0\}$$

Rank all unindexed nodes:

$$rank(v_1) = rank(v_2) = rank(v_3) = rank(v_4) = rank(v_5) = 0$$

Assign a temporary index to each node based on previous indexes (for indexed nodes) and ranks (for unindexed nodes):

$$\mathcal{I}^1(v_0) = 0, \mathcal{I}^1(v_1) = 1, \mathcal{I}^1(v_2) = 1, \mathcal{I}^1(v_3) = 1, \mathcal{I}^1(v_4) = 1, \mathcal{I}^1(v_5) = 1$$

After having indexes for all nodes, define temporary neighbor sets:

$$\mathcal{N}_t^1(v_1) = \{0, 1, 1, 1\}, \mathcal{N}_t^1(v_2) = \{0, 1, 1\}, \mathcal{N}_t^1(v_3) = \{0, 1, 1\},$$
$$\mathcal{N}_t^1(v_4) = \{0, 1, 1\}, \mathcal{N}_t^1(v_5) = \{0, 1\}$$

Based on the temporary neighbor sets, $v_1$ is chosen to be indexed 1 (i.e., $\mathcal{I}(v_1) = 1$).

$s = 2 : V_1^2 = \{v_0, v_1\}, V_2^2 = \{v_2, v_3, v_4, v_5\}$

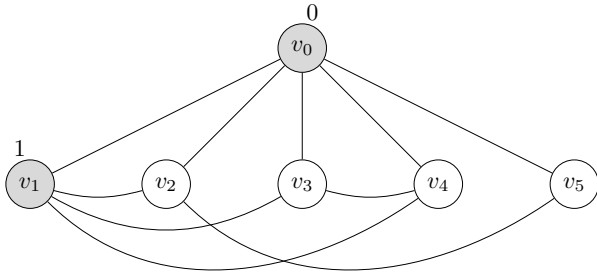

Figure 5: $V_1^2 = \{v_0, v_1\}$

Obtain indexed neighbors to each unindexed node:

$$\mathcal{N}^2(v_2) = \{0,1\}, \mathcal{N}^2(v_3) = \{0,1\}, \mathcal{N}^2(v_4) = \{0,1\}, \mathcal{N}^2(v_5) = \{0\}$$

Rank all unindexed nodes:

$$rank(v_2) = rank(v_3) = rank(v_4) = 0, rank(v_5) = 1$$

Assign a temporary index to each node based on previous indexes (for indexed nodes) and ranks (for unindexed nodes):

$$\mathcal{I}^2(v_2) = 2, \mathcal{I}^2(v_3) = 2, \mathcal{I}^2(v_4) = 2, \mathcal{I}^2(v_5) = 3$$

After having indexes for all nodes, define temporary neighbor sets:

$$\mathcal{N}_t^2(v_2) = \{0,1,3\}, \mathcal{N}_t^2(v_3) = \{0,1,2\}, \mathcal{N}_t^2(v_4) = \{0,1,2\}, \mathcal{N}_t^2(v_5) = \{0,2\}$$

Based on the temporary neighbor sets, both $v_3$ and $v_4$ can be chosen to be indexed 2. Without loss of generality, let $\mathcal{I}(v_3) = 2$.

**Note:** This step explains why temporary indexes and neighbor sets should be added into Algorithm 1. Otherwise, $v_2$ is also valid to be index 2, following which $v_3, v_4, v_5$ will be indexed $3, 4, 5$. Then the neighbor set for $\mathcal{I}(v_2) = 2$ is $\{0,1,5\}$ while the neighbor set for $\mathcal{I}(v_3) = 3$ is $\{0,1,4\}$, which violates constraints (S3).

$\boldsymbol{s = 3}:$ $V_1^3 = \{v_0, v_1, v_3\}, V_2^3 = \{v_2, v_4, v_5\}$

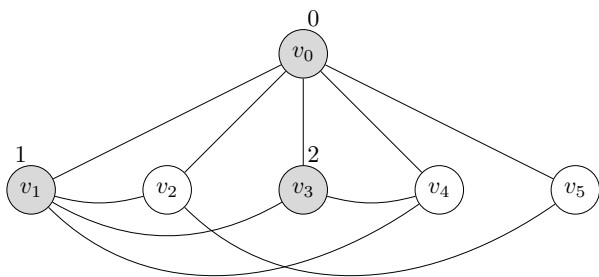

Figure 6: $V_1^3 = \{v_0, v_1, v_3\}$

Obtain indexed neighbors to each unindexed node:

$$\mathcal{N}^3(v_2) = \{0,1\}, \mathcal{N}^3(v_4) = \{0,1,2\}, \mathcal{N}^3(v_5) = \{0\}$$

Rank all unindexed nodes:

$$rank(v_2) = 1, rank(v_4) = 0, rank(v_5) = 2$$

Assign a temporary index to each node based on previous indexes (for indexed nodes) and ranks (for unindexed nodes):

$$\mathcal{I}^3(v_2) = 4, \mathcal{I}^3(v_4) = 3, \mathcal{I}^3(v_5) = 5$$

After having indexes for all nodes, define temporary neighbor sets:

$$\mathcal{N}_t^3(v_2) = \{0,1,5\}, \mathcal{N}_t^3(v_4) = \{0,1,2\}, \mathcal{N}_t^3(v_5) = \{0,4\}$$

Based on the temporary neighbor sets, $v_4$ is chosen to be indexed 3 (i.e., $\mathcal{I}(v_4) = 3$).

$\boldsymbol{s = 4}:$ $V_1^4 = \{v_0, v_1, v_3, v_4\}, V_2^4 = \{v_2, v_5\}$

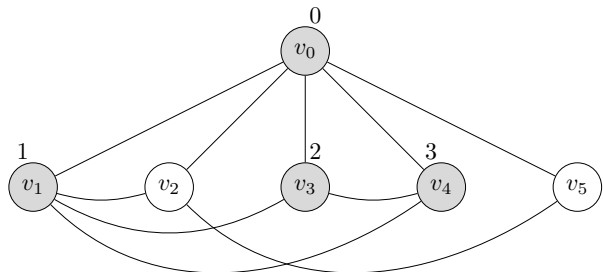

Figure 7: $V_1^4 = \{v_0, v_1, v_3, v_4\}$

Obtain indexed neighbors to each unindexed node:
$$\mathcal{N}^4(v_2) = \{0, 1\}, \mathcal{N}^4(v_5) = \{0\}$$
Rank all unindexed nodes:
$$rank(v_2) = 0, rank(v_5) = 1$$
Assign a temporary index to each node based on previous indexes (for indexed nodes) and ranks (for unindexed nodes):
$$\mathcal{I}^4(v_2) = 4, \mathcal{I}^4(v_5) = 5$$
After having indexes for all nodes, define temporary neighbor sets:
$$\mathcal{N}_t^4(v_2) = \{0, 1, 5\}, \mathcal{N}_t^4(v_5) = \{0, 4\}$$
Based on the temporary neighbor sets, $v_2$ is chosen to be indexed 4 (i.e., $\mathcal{I}(v_2) = 4$).

$s = 5 : V_1^5 = \{v_0, v_1, v_2, v_3, v_4\}, V_2^4 = \{v_5\}$

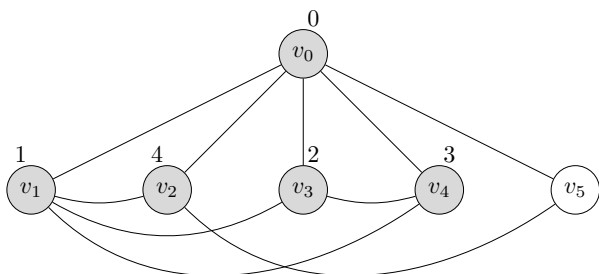

Figure 8: $V_1^5 = \{v_0, v_1, v_2, v_3, v_4\}$

Since there is only node $v_5$ unindexed, without running the algorithm we still know that $v_5$ is chosen to be indexed 5 (i.e., $\mathcal{I}(v_5) = 5$).

**Output:**

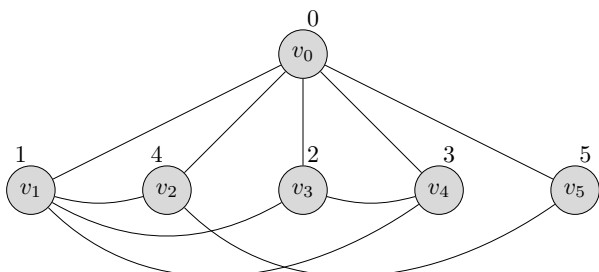

Figure 9: $V_1^6 = \{v_0, v_1, v_2, v_3, v_4, v_5\}$

# B  MIP formulation for CAMD: details

In this part, we provide details of the MIP formulation for CAMD, including parameters, variables and constraints. Especially, we use atom set $\{C, N, O, S\}$ as an example to help illuminate. The number of nodes $N$ exclude hydrogen atoms since they are implicitly represented by $N^h$ features.

## B.1  Parameters

Table 2: List of parameters

| Parameter | Description | Value |
|-----------|-------------|-------|
| $N$ | number of nodes | |
| $F$ | number of features | 16 |
| $N^t$ | number of atom types | 4 |
| $N^n$ | number of neighbors | 5 |
| $N^h$ | number of hydrogen | 5 |
| $I^t$ | index for $N^t$ | $\{0, 1, 2, 3\}$ |
| $I^n$ | index for $N^n$ | $\{4, 5, 6, 7, 8\}$ |
| $I^h$ | index for $N^h$ | $\{9, 10, 11, 12, 13\}$ |
| $I^{db}$ | index for double bond | 14 |
| $I^{tb}$ | index for triple bond | 15 |
| $Atom$ | atom types | $\{C, N, O, S\}$ |
| $Cov$ | covalence of atom | $\{4, 3, 2, 2\}$ |

## B.2  Variables

Table 3: List of variables for atom features

| $X_{v,f}$ | Type | $X_{v,f}$ | #Neighbors | $X_{v,f}$ | #Hydrogen |
|-----------|------|-----------|------------|-----------|-----------|
| 0 | C | 4 | 0 | 9 | 0 |
| 1 | N | 5 | 1 | 10 | 1 |
| 2 | O | 6 | 2 | 11 | 2 |
| 3 | S | 7 | 3 | 12 | 3 |
| | | 8 | 4 | 13 | 4 |

For the existence and types of bonds, we add the following extra features and variables:

- $X_{v, I^{db}}, v \in [N]$: if atom $v$ is included in at least one double bond.
- $X_{v, I^{tb}}, v \in [N]$: if atom $v$ is included in at least one triple bond.
- $A_{u,v}, u \neq v \in [N]$: if there is one bond between atom $u$ and $v$.
- $A_{v,v}, v \in [N]$: if node $v$ exists.
- $DB_{u,v}, u \neq v \in [N]$: if there is one double bond between atom $u$ and $v$.
- $TB_{u,v}, u \neq v \in [N]$: if there is one triple bond between atom $u$ and $v$.

*Remark:* Variables $A_{v,v}$ admits us to design molecules with at most $N$ atoms, which brings flexibility in real-world applications. In our experiments, however, we fix them to consider possible structures with exact $N$ atoms. Therefore, constraints (C2) are replaced by:

$$A_{v,v} = 1, \ \forall v \in [N]$$

Without this replacement, we can add an extra term on (C26) to exclude non-existing nodes and only index existing node:

$$\sum_{f \in [F]} 2^{F-f-1} \cdot X_{0,f} \leq \sum_{f \in [F]} 2^{F-f-1} \cdot X_{v,f} + 2^F \cdot (1 - A_{v,v}), \ \forall v \in [N] \backslash \{0\}$$

## B.3 Constraints

- There are at least two atoms and one bond between them:
$$A_{0,0} = A_{1,1} = A_{0,1} = 1 \tag{C1}$$

- Atoms with smaller indexes exist:
$$A_{v,v} \geq A_{v+1,v+1}, \ \forall v \in [N-1] \tag{C2}$$

- The adjacency matrix $A$ is symmetric:
$$A_{u,v} = A_{v,u}, \ \forall u,v \in [N], u < v \tag{C3}$$

- Atom $v$ is linked to other atoms if and only if it exists:
$$(N-1) \cdot A_{v,v} \geq \sum_{u \neq v} A_{u,v}, \ \forall v \in [N] \tag{C4}$$

*Note:* $N - 1$ here is a big-M constant.

- If atom $v$ exists, then it has to be linked with at least one atom with smaller index:
$$A_{v,v} \leq \sum_{u < v} A_{u,v}, \ \forall v \in [N] \tag{C5}$$

- Atom $v$ cannot have a double bond with it self:
$$DB_{v,v} = 0, \ \forall v \in [N] \tag{C6}$$

- DB is also symmetric:
$$DB_{u,v} = DB_{v,u}, \ \forall u,v \in [N], u < v \tag{C7}$$

- Atom $v$ cannot have a triple bond with it self:
$$TB_{v,v} = 0, \ \forall v \in [N] \tag{C8}$$

- TB is also symmetric:
$$TB_{u,v} = TB_{v,u}, \ \forall u,v \in [N], u < v \tag{C9}$$

- There is at most one bond type between two atoms:
$$DB_{u,v} + TB_{u,v} \leq A_{u,v}, \ \forall u,v \in [N], u < v \tag{C10}$$

- If atom $v$ exists, then it has only one type:
$$A_{v,v} = \sum_{f \in I^t} X_{v,f}, \ \forall v \in [N] \tag{C11}$$

- If atom $v$ exists, then it has only number of neighbors:
$$A_{v,v} = \sum_{f \in I^n} X_{v,f}, \ \forall v \in [N] \tag{C12}$$

- If atom $v$ exists, then it has only one number of hydrogen:
$$A_{v,v} = \sum_{f \in I^h} X_{v,f}, \ \forall v \in [N] \tag{C13}$$

- The number of neighbors of atom $v$ equals to its degree:
$$\sum_{u \neq v} A_{u,v} = \sum_{i \in [N^n]} i \cdot X_{v,I_i^n}, \ \forall v \in [N] \tag{C14}$$

- If there is possibly one double bond between atom $u$ and $v$, then the double bond feature of both atoms should be 1 and these two atoms are linked:
$$3 \cdot DB_{u,v} \leq X_{u,I^{db}} + X_{v,I^{db}} + A_{u,v}, \ \forall u,v \in [N], u < v \tag{C15}$$

- If there is possibly one triple bond between atom $u$ and $v$, then the triple bond feature of both atoms should be 1 and these two atoms are linked:

$$3 \cdot TB_{u,v} \leq X_{u,I^{tb}} + X_{v,I^{tb}} + A_{u,v}, \ \forall u, v \in [N], u < v \qquad \text{(C16)}$$

- The maximal number of double bonds for atom $v$ is limited by its covalence:

$$\sum_{u \in [N]} DB_{u,v} \leq \sum_{i \in [N^t]} \left\lfloor \frac{Cov_i}{2} \right\rfloor \cdot X_{v,I_i^t}, \ \forall v \in [N] \qquad \text{(C17)}$$

- The maximal number of triple bonds for atom $v$ is limited by its covalence:

$$\sum_{u \in [N]} TB_{u,v} \leq \sum_{i \in [N^t]} \left\lfloor \frac{Cov_i}{3} \right\rfloor \cdot X_{v,I_i^t}, \ \forall v \in [N] \qquad \text{(C18)}$$

- If atom $v$ is not linked with other atoms with double bond, then its double bond feature should be 0:

$$X_{v,I^{db}} \leq \sum_{u \in [N]} DB_{u,v}, \ \forall v \in [N] \qquad \text{(C19)}$$

- If atom $v$ is not linked with other atoms with triple bond, then its triple bond feature should be 0:

$$X_{v,I^{tb}} \leq \sum_{u \in [N]} TB_{u,v}, \ \forall v \in [N] \qquad \text{(C20)}$$

- The covalence of atom $v$ equals to the sum of its neighbors, hydrogen, double bonds, and triple bonds:

$$\sum_{i \in [N^t]} Cov_i \cdot X_{v,I_i^t} = \sum_{i \in [N^n]} i \cdot X_{v,I_i^n} + \sum_{i \in [N^h]} i \cdot X_{i,I_i^h}$$
$$+ \sum_{u \in [N]} DB_{u,v} + \sum_{u \in [N]} 2 \cdot TB_{u,v}, \ \forall v \in [N] \qquad \text{(C21)}$$

- Practically, there could be bounds for each type of atom:

$$LB_{Atom_i} \leq \sum_{v \in [N]} X_{v,I_i^t} \leq UB_{Atom_i}, \ \forall i \in [N^t] \qquad \text{(C22)}$$

- Practically, there could be bounds for number of double bonds:

$$LB_{db} \leq \sum_{v \in [N]} \sum_{u < v} DB_{u,v} \leq UB_{db} \qquad \text{(C23)}$$

- Practically, there could be bounds for number of triple bonds:

$$LB_{tb} \leq \sum_{v \in [N]} \sum_{u < v} TB_{u,v} \leq UB_{tb} \qquad \text{(C24)}$$

- Practically, there could be bounds for number of rings. Since we force the connectivity, $N - 1$ bonds are needed to form a tree and then each extra bond forms a ring. Therefore, we can bound the number of rings by:

$$LB_{ring} \leq \sum_{v \in [N]} \sum_{u < v} A_{u,v} - (N - 1) \leq UB_{ring} \qquad \text{(C25)}$$

## C Implementation details and more results

### C.1 Dataset preparation

Both datasets used in this paper are available in python package *chemprop* [38]. Molecular features (133 atom features and 14 bond features) are extracted from their SMILES representations. Based on the these features and our definition for variables in Appendix B.2, we calculate $F = 16$ (since both datasets have 4 types of atoms) features for each atom, construct adjacency matrix $A$ as well as double and triple bond matrices $DB$ and $TB$. In this part, we first briefly introduce these datasets and then provide details about how to preprocess them.

**QM7 dataset:** Consist of $6,830$ molecules with at most 7 heavy atoms $C, N, O, S$. One quantum mechanics property is provided for each molecule. We filter the dataset and remove three types of incompatible molecules: (1) molecules with aromaticity (i.e., with float electron(s) inside a ring); (2) molecules constraints $S$ atom with covalence 6 (since one type of atom can only have one covalence in our formulation); (3) molecules with non-zero radical electrons. After preprocessing, there are $5,822$ molecules left with properties re-scaled to $0 \sim 1$.

To calculate the bounds needed in constraints (C22) - (C25), we check all molecules in QM7 dataset. The maximal ratios between the number of N,O and S atoms and the total atoms are $3/7, 1/3, 1/7$, respectively, while the minimal ratio between the number of C atoms and the total atoms is $1/2$. The maximal ratios between the number of double bonds, triple bonds, rings and the number of atoms are $3/7, 3/7, 3/7$, respectively. Therefore, we set the following bounds:

$$LB_C = \left\lceil \frac{N}{2} \right\rceil, UB_N = \max\left(1, \left\lfloor \frac{3N}{7} \right\rfloor\right), UB_O = \max\left(1, \left\lfloor \frac{N}{3} \right\rfloor\right), UB_S = \max\left(1, \left\lfloor \frac{N}{7} \right\rfloor\right)$$
$$UB_{db} = \left\lfloor \frac{N}{2} \right\rfloor, UB_{tb} = \left\lfloor \frac{N}{2} \right\rfloor, UB_{ring} = \left\lfloor \frac{N}{2} \right\rfloor$$

**QM9 dataset:** Consist of $133,885$ molecules with at most 9 heavy atoms $C, N, O, F$. 12 quantum mechanics properties are provided for each molecule. We filter the dataset and remove two types of incompatible molecules: (1) molecules with aromaticity; (2) molecules with non-zero radical electrons. After preprocessing, there are $108,723$ molecules left. Among these features, we choose one property $U_0$, which denotes the internal energy at 0K. These properties are re-scaled to $0 \sim 1$.

Similarly, we set the following bounds for the experiments on the QM9 dataset:

$$LB_C = \left\lceil \frac{N}{5} \right\rceil, UB_N = \left\lfloor \frac{3N}{5} \right\rfloor, UB_O = \left\lfloor \frac{4N}{7} \right\rfloor, UB_F = \left\lfloor \frac{4N}{5} \right\rfloor$$
$$UB_{db} = \left\lfloor \frac{N}{2} \right\rfloor, UB_{tb} = \left\lfloor \frac{N}{2} \right\rfloor, UB_{ring} = \left\lfloor \frac{2N}{3} \right\rfloor$$

### C.2 Training of GNNs

For QM7 dataset, the GNN's structure is shown in Table 4. For each run, we first randomly shuffle the dataset, then use the first 5000 elements to train and the last 822 elements to test. Each model is trained for 100 iterations with learning rate 0.01 and batch size 64. The average training ($l_1$) loss is 0.0241 and test loss is 0.0285.

Table 4: Structure of GNN for QM7 dataset

| index | layer | in features | out features | activation |
|-------|-------|-------------|--------------|------------|
| 1 | SAGEConv | 16 | 16 | ReLU |
| 2 | SAGEConv | 16 | 32 | ReLU |
| 3 | Pooling | $32N$ | 32 | None |
| 4 | Linear | 32 | 16 | ReLU |
| 5 | Linear | 16 | 4 | ReLU |
| 6 | Linear | 4 | 1 | None |

For QM9 dataset, the GNN's structure is shown in Table 5. We apply the same experimental set-up as before, but with a larger number of channels to fit this much larger dataset. For each run, after randomly shuffled, the first 80000 elements are used to train and the last 28723 elements are used to test. The average training loss is 0.0036 and test loss is 0.0036.

Table 5: Structure of GNN for QM9 dataset

| index | layer | in features | out features | activation |
|-------|-------|-------------|--------------|------------|
| 1 | SAGEConv | 16 | 32 | ReLU |
| 2 | SAGEConv | 32 | 64 | ReLU |
| 3 | Pooling | $64N$ | 64 | None |
| 4 | Linear | 64 | 16 | ReLU |
| 5 | Linear | 16 | 4 | ReLU |
| 6 | Linear | 4 | 1 | None |

For both datasets, the trained GNNs achieve comparable accuracy to the state-of-the-art methods [38]. Note that this comparison is unfair since we preprocess each dataset and remove some incompatible molecules. Also, we only choose one property to train and then optimize. However, low losses mean that the trained GNNs approximate true properties well.

## C.3 Full experimental results

We report full experimental results in this section. For each dataset with a given $N$ and a formulation, numbers of variables ($\#var_c$: continuous, $\#var_b$: binary) and constraints ($\#con$) after presolve stage in Gurobi are first reported. Then we count the number of successful runs (i.e., achieving optimality) $\#run$. For all successful runs, we provide the mean, 25th percentile ($Q_1$), and 75th percentilie ($Q_3$) of the running time $time_{tol}$ as well as the first time to find the optimal solution $time_{opt}$. The time limit for each run is 10 hours. Except for the time limit and random seed, we use the default setting in Gurobi for other parameters such as the tolerance.

Table 6: Numerical results for QM7 dataset.

| $N$ | method | $\#var_c$ | $\#var_b$ | $\#con$ | $\#run$ | $time_{tol}$ (s) mean | $Q_1$ | $Q_3$ | $time_{opt}$ (s) mean | $Q_1$ | $Q_3$ |
|---|---|---|---|---|---|---|---|---|---|---|---|
| 4 | bilinear | **616** | 411 | 1812 | 50 | 477 | 314 | 579 | 101 | **13** | 147 |
| | bilinear+BrS | **616** | 383 | 1755 | 50 | 149 | 115 | 167 | 94 | 66 | 131 |
| | big-M | 792 | 281 | 1838 | 50 | 514 | 206 | 722 | 130 | 15 | 150 |
| | big-M+BrS | 760 | **269** | **1748** | 50 | **114** | **91** | **128** | **53** | **13** | **94** |
| 5 | bilinear | **840** | 611 | 2776 | 37 | 9365 | 3551 | 9029 | 471 | 169 | 339 |
| | bilinear+BrS | **840** | 565 | 2679 | 50 | 659 | 282 | **652** | **198** | 136 | **257** |
| | big-M | 1142 | 359 | 2795 | 50 | 6369 | 1424 | 9703 | 729 | **74** | 616 |
| | big-M+BrS | 1097 | **346** | **2660** | 50 | **501** | **266** | 660 | 205 | 91 | 281 |
| 6 | bilinear | 1096 | 828 | 3893 | 2 | 23853 | 22081 | 25627 | **389** | 373 | **396** |
| | bilinear+BrS | **1064** | 747 | **3602** | 48 | 6400 | 1748 | 6225 | 1751 | **334** | 2154 |
| | big-M | 1628 | 436 | 4018 | 10 | 21484 | 16095 | 24848 | 14368 | 9966 | 19930 |
| | big-M+BrS | 1505 | **420** | 3665 | 47 | **4700** | **1656** | **3759** | 1156 | 515 | 1440 |
| 7 | bilinear | 1384 | 1076 | 5193 | 0 | * | * | * | * | * | * |
| | bilinear+BrS | **1320** | 950 | **4684** | 47 | 12989 | 4651 | 19229 | **3366** | 497 | **4587** |
| | big-M | 2134 | 516 | 5364 | 0 | * | * | * | * | * | * |
| | big-M+BrS | 1943 | **495** | 4806 | 49 | **10802** | **4159** | **16101** | 4008 | **64** | 5251 |
| 8 | bilinear | 1704 | 1355 | 6676 | 0 | * | * | * | * | * | * |
| | bilinear+BrS | **1608** | 1181 | **5941** | 12 | 16219 | 7623 | 22951 | 3082 | 90 | 223 |
| | big-M | 2676 | 599 | 6869 | 0 | * | * | * | * | * | * |
| | big-M+BrS | 2390 | **572** | 6040 | 12 | **6606** | **2132** | **3120** | 2854 | **55** | **87** |

Table 7: Numerical results for QM9 dataset.

| $N$ | method | $\#var_c$ | $\#var_b$ | $\#con$ | $\#run$ | $time_{tol}$ (s) | | | $time_{opt}$ (s) | | |
|---|---|---|---|---|---|---|---|---|---|---|---|
| | | | | | | mean | $Q_1$ | $Q_3$ | mean | $Q_1$ | $Q_3$ |
| 4 | bilinear | **1178** | 599 | 3050 | 50 | 368 | 252 | 353 | 240 | 186 | 279 |
| | bilinear+BrS | 1181 | 580 | 3020 | 50 | 256 | 211 | 288 | 184 | 60 | 258 |
| | big-M | 1340 | 453 | 2933 | 50 | 258 | 193 | 311 | 190 | 141 | 235 |
| | big-M+BrS | 1316 | **441** | **2863** | 50 | **167** | **89** | **227** | **111** | **43** | **171** |
| 5 | bilinear | **1617** | 846 | 4563 | 48 | 5439 | 1108 | 7573 | 2604 | 585 | 1494 |
| | bilinear+BrS | 1627 | 823 | 4545 | 50 | **1039** | 704 | **1248** | **628** | 579 | **744** |
| | big-M | 1962 | 584 | 4480 | 50 | 3242 | 1111 | 3535 | 1642 | 757 | 1762 |
| | big-M+BrS | 1877 | **564** | **4260** | 50 | 1127 | **672** | 1336 | 640 | **397** | 868 |
| 6 | bilinear | 2118 | 1110 | 6319 | 21 | 16767 | 7598 | 25642 | 2969 | 479 | 1343 |
| | bilinear+BrS | **2068** | 1049 | 6017 | 48 | 6907 | 3595 | 8691 | 2394 | 491 | 1937 |
| | big-M | 2768 | 718 | 6562 | 29 | 12625 | 5573 | 18020 | 3626 | 314 | 2181 |
| | big-M+BrS | 2500 | **684** | **5829** | 50 | **1985** | **660** | **2613** | 1132 | **302** | 1243 |
| 7 | bilinear | 2681 | 1405 | 8348 | 3 | 22759 | 19543 | 24667 | **1417** | 995 | 1722 |
| | bilinear+BrS | **2571** | 1308 | 7768 | 34 | 21649 | 16723 | 27748 | 2983 | 813 | 3808 |
| | big-M | 3549 | 846 | 8649 | 1 | **6210** | 6210 | **6210** | 6108 | 6108 | 6108 |
| | big-M+BrS | 3180 | **810** | **7613** | 49 | 6586 | **2946** | 9655 | 1427 | **564** | **1255** |
| 8 | bilinear | 3306 | 1731 | 10650 | 0 | $*$ | $*$ | $*$ | $*$ | $*$ | $*$ |
| | bilinear+BrS | **3136** | 1597 | 9790 | 0 | $*$ | $*$ | $*$ | $*$ | $*$ | $*$ |
| | big-M | 4420 | 977 | 11013 | 0 | $*$ | $*$ | $*$ | $*$ | $*$ | $*$ |
| | big-M+BrS | 3922 | **935** | **9595** | 29 | **17864** | **11113** | **21431** | **3405** | **705** | **4308** |

