# OpenReview forum: "Optimizing over trained GNNs via symmetry breaking"
_NeurIPS.cc/2023/Conference — NeurIPS 2023 poster_

### Official Review · Reviewer_eLVB · 2023-06-18

**Soundness:** 3 good
**Presentation:** 2 fair
**Contribution:** 3 good
**Rating:** 6
**Confidence:** 2

**Summary:**

To overcome the symmetry issue when solving inverse problems on trained GNNs, this paper proposes two types of symmetry-breaking constraints to break symmetry. The authors construct an indexing algorithm, and prove that the resulting graph indexing satisfies the proposed symmetry-breaking constraints.  They also develop two mixed-integer optimization formulations in the case where the input
graph is not fixed.

**Strengths:**

1. The symmetry issue which this paper aims to overcome is important, and the proposed indexing is a general approach that applies in many kinds of situations.
2. The theoretical analysis is solid, and the proof seems to be correct.

**Weaknesses:**

1. The experiments are not persuasive enough. The reasons are as follows.

(1) For Sec 3.1 (Mixed-integer optimization formulation for molecular design): In my opinion, a formulation itself WITHOUT experimental verification should not be thought of as a contribution.

(2) For Sec 3.2 (Counting feasible structures: performance of symmetry breaking): The experimental results (Table 1) do not have baselines.

2. There is no related work analysis (except a little description in Introduction). That makes it a little hard to justify the originality. For example, “The second formulation generalizes the big-M formulation for GraphSAGE in [70]” (L197). However, the authors did not explain how the big-M formulation looks like, in which way the proposed formulation generalizes it, and are there overlaps between the constraints of the big-M formulation and the proposed ones?

**Questions:**

1. I suggest that the authors provide complexity analysis to the proposed algorithm, since in Table 1 there are many time out cases.

2. I suggest that the authors discuss the relationship between the proposed algorithms and related works (please see Weaknesses (2)).

3. I suggest that the authors provide necessary experiments (please see Weaknesses (1)).

**Limitations:**

Yes.

---

> ### Author Rebuttal · Authors · 2023-08-08
>
> Sincerely appreciate these comments and questions.
>
> **Weakness 1(1) [Experimental verification of molecular constraints]**\
> Section 3.1 is not a contribution of our paper but a description of how we followed the literature. This section just provides background for our numerical experiments.
>
> We don't experimentally verify the Section 3.1 constraints because these are not new in the CAMD literature. On line 218 of Section 3.1, we explain the source of constraints C1 - C21 [70]. These constraints C1 - C21 are very similar to those which have been in the literature since 1993 [92].
>
> The new constraints that we add are C22 - C25. The reviewer makes a good point that we should justify these constraints and provide an explanation where these constraints appear. These constraints bound the number of atoms, bonds, and rings. We calculated these bounds based on the QM7 and QM9 data sets, so every molecule in the QM7 and QM9 database will satisfy these bounds, respectively. Appendix C.1 gives more details as to how these bounds were calculated, so we will also provide a link to Appendix C.1 for C22 - C25. The reason that we do not experimentally verify these new molecular constraints is that removing the constraints would allow the GNN to extrapolate to molecules dissimilar to its dataset, for example removing the bound on the number of oxygen atoms could allow a molecule with 7 or 9 atoms to have 4 oxygen atoms when there is no example in the database that has 4 oxygen atoms.
>
> **Weakness 1(2) \& Question 3 [Baseline of Table 1]**\
> Thanks for pointing this out. The *(S1)* column in Table 1 is the baseline, but we will clarify this in both Section 3.2 and the caption of Table 1. The reason that the *(S1)* column is the baseline is that these constraints come from the experimentally-verified literature and force the connectivity of molecules.
>
> **Weakness 2 \& Question 2 [Originality of big-M formulation]**
>
> It's a good point that we should more clearly explain how we generalize this big-M formulation. Currently (and we will make this more clear), the formulation in lines 197 - 204 is exactly the same as [70] except that we use our notation to keep consistency.
>
> The way we generalize this formulation is not by changing these mathematical equations but rather by showing that this formulation is not limited to GraphSAGE. The mathematical formulation in lines 197 - 204 for all GNN architectures satisfying equation (*) in line 181 and the assumptions in line 191. We agree that the link from lines 181 \& 191 to lines 197 - 204 are not clear, so we will make the connection explicit in Section 2.5.2.
>
> **Question 1 [Complexity analysis]**\
> Table 1 counts all feasible solutions after adding more and more symmetry-breaking constraints. Since it's very hard to directly calculate these numbers analytically, we use MIP solver Gurobi to search the branch-and-bound tree to find every feasible solution. This process is super time-consuming. Our MIP for CAMD only involves binary variables, the maximum number of branches is $2^B$, where $B$ is the number of binary variables. That's why there are many time outs in Table 1: Gurobi can not visit all branches within the time limit (i.e., 48 hours).

---

> > ### Comment · Reviewer_eLVB · 2023-08-12
> > **Response to Authors**
> >
> > Thanks for your reply. You have already solved my concerns, so that I improve the score.

---

> > > ### Author Response · Authors · 2023-08-12
> > >
> > > We really appreciate that! Thanks again for your valuable comments.

---

### Official Review · Reviewer_5u3v · 2023-06-24

**Soundness:** 3 good
**Presentation:** 3 good
**Contribution:** 3 good
**Rating:** 7
**Confidence:** 3

**Summary:**

This paper investigates the optimization of trained GNNs. This is a permutation-invariant problem and all points on an orbit of the symmetric group have the same performance. The authors proposed some symmetry-breaking approaches to have smaller search space and hence less redundancy and more efficient optimization. The correctness is proved theoretically and the efficiency is examined numerically.

**Strengths:**

1. Optimizing a trained GNN is an important topic in practice and breaking symmetry can reduce the redundancy in many optimization problems.
2. The authors provide the theoretical guarantee of the proposed algorithm.
3. The numerical results look nice. It is reasonable to expect such improvement since the MIP formulation does have large symmetry group.

**Weaknesses:**

1. I do not think this paper has enough literature review on detecting and breaking MIP symmetry -- how do previous work detect symmetry? what is the cost for symmetry detection? what is the connection/difference between previous symmetry-breaking approaches and the proposed approach? I think Gurobi can detect symmetry but the computational cost might depend on the size of the symmetry group. For optimizing GNNs, the symmetry group is obvious (just the permutation group). So one does not need to detect the symmetry and can design problem-specific symmetry-breaking approaches. Those ideas/background should be discussed more in the paper.
2. As the authors discuss in Section 4, the numerical experiments are not general enough, but I think the current experiments have already shown that the approach is promising.

**Questions:**

If you have Gurobi to detect and break the symmetry, will Gurobi add any constraints and if so, are those constraints the same as or different from (C26) and (C27)?

**Limitations:**

The authors clearly discuss the limitations in Section 4 and I agree with their discussion.

---

> ### Author Rebuttal · Authors · 2023-08-08
>
> Thank you for these considerations about the relationship and difference between our work and the literature.
>
> **Weakness [Literature review]**\
> Sincere thanks for this useful comment on the literature review: indeed we will add more details in the final paper. MIP solvers typically detect symmetry using graph automorphism, for example the open-source SCIP solver uses BLISS [i, ii], and then both Gurobi and SCIP break symmetry using orbital fixing [iii] and orbital pruning [iv].
>
> The question about the cost of symmetry-breaking is great because it helps explain why our symmetry-breaking strategy is working so well. When a MIP solver detects symmetry, the only graph it has available is the graph formed by the variables and constraints in the MIP. So, for instance when $N = 4$, we only need to consider a permutation group with $N!=24$ elements. However, because the MIP solver does not have access to the graph structure, it needs to consider all possible automorphic graphs with $616+411$ nodes (these are the number of continuous and binary variables in the first line of Table 6 in our paper). So while Gurobi and the other MIP solvers *can* detect and break the symmetry, the computational cost of doing so is very high. We are reducing the computational cost for the solver by eliminating the need to look for these symmetries.
>
> `[i] Junttila, Tommi and Kaski, Petteri. Conflict propagation and component recursion for canonical labeling. TAPAS, 2011.`\
> `[ii] Junttila, Tommi and Kaski, Petteri. Engineering an efficient canonical labeling tool for large and sparse graphs. ALENEX, 2007.`\
> `[iii] Kaibel, Volker and Peinhardt, Matthias and Pfetsch, Marc E. Orbitopal fixing. Discrete Optimization, 2011.`\
> `[iv] Margot, François. Pruning by isomorphism in branch-and-cut. Mathematical Programming, 2002.`
>
> **Weakness [Connection \& Difference to the literature]**\
> We have (an admittedly short) answer to this question in lines 169 - 173 of Section 2.4. However, we will make this clearer by adding some details.
>
> The most similar setting to ours is an application that distributes $m$ different jobs to $n$ identical machines and minimizes the total cost. The requirement is that each job can only be assigned to one machine (but each machine can be assigned to more than one job). Symmetry comes from all permutations of machines while variables $A_{i,j}$ denote if job $i$ is assigned to machine $j$. This setting appears in noise dosage problems [83, 84], packing and partitioning orbitopes [85, 86], and scheduling problems [87]. However, requiring that the sum of each row in $A_{i,j}$ equals to $1$ simplifies the problem. By forcing decreasing lexicographical orders for all columns, the symmetry issue is handled well. Our proposed constraints can be regarded as a non-trivial generalization of these constraints from a bipartite graph to an arbitrary undirected graph.
>
> The main difference between the symmetry issue in this paper versus in the literature is that our variables are features and an adjacency matrix while the symmetry comes from alternative indexing of abstract nodes (e.g., atoms). As the reviewer mentioned, the symmetry group is the permutation group. Any element in the symmetry group acts on $N$ nodes to produce a isomorphic graph in graph level. In MIP, however, it corresponds to an element of symmetry group for all variables, including node features ($O(NF)$), adjacency matrix ($O(N^2)$), model (e.g., GNN), and problem-specific features (e.g., double bonds $DB$). By adding the proposed symmetry-breaking constraints, we don't need to consider a much larger symmetry group defined on all variables to find a subgroup which corresponds to the true symmetric group: the permutation group defined on abstract nodes.

---

> > ### Comment · Reviewer_5u3v · 2023-08-13
> >
> > Thank you very much for your reply. The discussion about the previous literature and connection/difference look very nice.

---

> > > ### Author Response · Authors · 2023-08-14
> > >
> > > We are glad to hear that. Many thanks for pointing our lack of this literature!

---

### Official Review · Reviewer_ni19 · 2023-07-06

**Soundness:** 3 good
**Presentation:** 4 excellent
**Contribution:** 3 good
**Rating:** 7
**Confidence:** 3

**Summary:**

This paper suggests using a symmetry breaking indexing for nodes in a graph. The argument is that for inverse problems, such as molecule design, graph isomorphism results in finding many equivalent solutions. The symmetry breaking indices are supposed to partially alleviate this redundancy. They use pre-trained GNN (based on GraphSage) for computer-aided molecule design (CAMD). During the design, they use Mixed-Integer Programming (MIP) (solved using Gurobi) to solve a set of constraints to make the molecules chemically realistic as well as constraints arising from their indexing scheme.

They conduct experiments on QM7 and QM9 and find that the symmetry breaking methods perform much better. They show more successful runs (i.e. producing molecules satisfying the constraints) and achieve this in a fraction of the time, compared to models without symmetry breaking.

**Strengths:**

The paper makes a good case for how adding an ordering (indexing nodes in a specific manner) can dramatically improve inverse problems on graphs, e.g. designing molecules, solved using MIP.
The paper is well structured and discusses the literature in good depth. It explains the algorithm and experimental setup. The results are also very strong for the experiments.

**Weaknesses:**

The experiments show that with the indexing, MIP converges much faster. Table 1 states that MIP find many feasible solutions for each setting and set of constraints. One thing I hope the authors can assess (maybe using a Weisfeiler-Lehman hash or similar; see questions below) is how much the symmetry breaking *actually* removes isomorphic solutions, compared to the runs without symmetry breaking. they could do it for a small setting, maybe QM9, N=3 or 4. The goal would be two fold:
1. To show that the symmetry breaking indeed removed many isomorphic molecules
2. to check how the diversity of solutions in the symmetry broken case compares with the diversity of molecule found in the vanilla algorithm.

I was also hoping some kind of baseline could be discussed, but I understand that to have a fair comparison they need to use the same pretrained GNNs with and without symmetry breaking.

Also, the claims about generality of the approach and applicability to other graph problems, though reasonable, lack any substantive evidence or hint as to how to do it concretely. I suggest wither softening the claims or giving some hints as to why that is.

In terms of structure, I think the proof of theorem 1 is not necessary to be in the main text and occupies too much space. Instead some version of the last two tables in the appendix could be more useful in the main text.

**Questions:**

1. Table 1: In each row, is higher necessarily better? If we want less symmetry copies, should sometimes less be better?
2. How do we know that the solutions found with the symmetry broken model are as diverse as the one with symmetries? These are small molecules. Can you use a fast Weisfeiler-Lehman type hash to check isomorphism of found molecules at least in small N experiments? Maybe also use a measure of diversity, some kind of entropy or something.

### Minor questions:
line 75 “Directed to undirected” how?
line 106 A is now a multiset, not the adjacency matrix anymore?
line 183 (*) is not permutation invariant, as written. $w_{u\to v}$ is like a generalized Graph Attention (GAT).
line 187 Dense NN: do you mean a fully connected layer (no graph)?

**Limitations:**

The authors discuss technical limitations, but I don't think there are immediate societal impacts, as the paper pertains to a technical method to speed up generative design in graph problems.

---

> ### Author Rebuttal · Authors · 2023-08-08
>
> Many thanks for these helpful comments and suggestions.
>
> **Weakness [How many symmetric solutions are removed]**\
> We acknowledge that the limited information in Section 3.2 and Table 1 is insufficient for our purposes. Table 1 is not used to show that MIP can find *many* feasible solutions under different settings. These experiments numerically count *all* feasible solutions based on different levels of symmetry-breaking constraints.
>
> In Table 1, the *(S1)* column is the baseline since each molecule should at least be a connected graph. Constraints S1 (which force connectivity) come from the literature and we wanted to show the improvement after adding our proposed symmetry-breaking constraints S2 and S3 with respect to S1. Therefore, the differences between the *(S1)* column and the *(S1)-(S3)* column are the number of removed symmetric solutions. Note that we did not use *symmetric molecules* as the reviewer mentioned. Please see our comments to your question about diversity of solutions.
>
> The *(S1)* column is our baseline because, without adding S1, the baseline is the number of any graph with compatible features. If we ignore all features and only consider all possible graph structures, the baseline is $2^{\frac{N(N-1)}{2}}$. This number will be much larger after introducing features, which loses meaning as a baseline.
>
> The final paper will provide more information about the purpose of Section 3.2 and clarify the baseline in Table 1. We will add one more column for each dataset in Table 1 to show the percentage of removed symmetric solutions after adding S2 and S3.
>
> **Weakness [Generality consideration]**\
> Thanks for this great suggestion. Due to space limitation, we only briefly discussed the generality in Section 2.4 and Section 4. We provide some ideas here and will enrich the relevant discussion in the final paper to include these ideas.
>
> - The realization of these constraints is not limited to MIP: they also can exclude symmetric solutions in other optimization methods, such as genetic algorithms.
>
> - The constraints are independent of GNN: GNN in our work is just a permutation-invariant function. Our constraints directly work on graphs and features. In traditional computer-aided molecular design (CAMD), for example, the score functions are artificially-designed with closed forms. Since these functions are commonly-defined on features of molecules, they are also permutation-invariant. Similarly, other ML models could replace GNNs as long as they are permutation-invariant. Therefore, CAMD *without* a GNN is another possible application of our work.
>
> As recommended by the reviewer, we will weaken the statements about the generality of the applications. The applications that we know of for this work are (i) GNN, (ii) CAMD, and (iii) the scheduling instances mentioned in Section 2.4. However, the statement of applications beyond these three is an aspirational one and we do not concretely know of other applications.
>
> **Weakness [Paper structure]**\
> We appreciate the concern about the paper structure. Since there will be one more page in final paper, we will try to add a simplified version of last two tables to the main text. We’re hoping that we can get both the proof and the tables into the final version (while also addressing the other promised changes).
>
> **Question [Diversity of solutions]**\
> In Table 1, fewer is always better, because it means that more symmetries are removed. Adding symmetry-breaking constraints will not influence the diversity of solutions. That is why we need to prove that there is at least one feasible indexing for each molecule. For example, if there are $n$ possible molecules in the design space, there are far more than $n$ feasible solutions due to symmetry, that is, multiple solutions correspond to the same molecule. The target of symmetry-breaking is to make the number of solutions as close as possible to $n$. No matter how many symmetries are removed, the diversity of solutions will not change because each molecule corresponds to at least one solution.
>
> **Minor questions**
> - line 75: We can just ignore the direction of each edge to transform a directed graph to an undirected graph.
>
> - line 106: Apologies for the ambiguity. $A$ is used in the definition of $LO(\cdot)$ and the theoretical part to denote a multiset. After introducing constraints S1, there is no need of the adjacency matrix. Except for lines 106 - 126 and the proof of Property 3 (Section A.1), $A$ is always the adjacency matrix. In final paper, we will either clarify this point before line 106, or change the notation for multisets.
>
> - line 183: For a mixed-integer optimization formulation of GNNs, we do not require that the GNN is permutation invariant, but only need that the GNN could be rewritten in form (*). To optimize over a trained GNN with a non-fixed graph structure, we require permutation invariance. If we wanted to support GAT, then the symmetry-breaking constraints would be incorrect since different indexing would correspond different GAT outputs.
>
> - line 187: Yes, Dense NN is used to denote a neural network consist of fully connected layers (or to say, dense layers, linear layers). We will clarify this in the final paper.

---

> > ### Comment · Reviewer_ni19 · 2023-08-16
> > **Thank you**
> >
> > Thank you for the clarifications about Table 1 and the rest of the paper. The diversity comment also makes sense, I think.

---

> > > ### Author Response · Authors · 2023-08-16
> > >
> > > Many thanks for your comments! We will improve our paper based on these clarifications.

---

### Official Review · Reviewer_eveB · 2023-07-07

**Soundness:** 3 good
**Presentation:** 4 excellent
**Contribution:** 3 good
**Rating:** 7
**Confidence:** 3

**Summary:**

This paper presents novel symmetry-breaking constraints for optimizing trained graph neural networks (GNNs) and addresses the graph isomorphism issue. The authors develop two mixed-integer optimization formulations and evaluate their methods in the context of molecular design.

**Strengths:**

1. **Relevance and Application**: The paper addresses the important issue of GNN optimization and demonstrates practical utility through molecular design application.

2. **Empirical Evidence**: The authors provide convincing experimental results supporting their claims, underscoring the proposed method's effectiveness.

3. **Clarity**: The manuscript is well-written, presenting complex ideas with notable clarity.


**Weaknesses:**

1. **Insufficient exploration of S1 and S3 compatibility**: The paper falls short in exploring whether S1 and S3 requirements can coexist for certain indexings of a connected graph with features.

2. **Insufficient explanation of structure feasibility constraints**: There is a lack of clarity concerning the interplay between structure feasibility constraints C1 - C21 and the indexing process. Specifically, the paper does not elucidate the number of these constraints that are related to indexing and if they are compatible with S3.


**Questions:**

1. Do the indexing produced by Algorithm 1 for a connected graph satisfy S1?

---

> ### Author Rebuttal · Authors · 2023-08-08
>
> Sincere thanks for these comments which are fair and help us clarify and improve our work.
>
> **Weakness 1 \& Question 1 [Compatibility of S1 and S3]**\
> Thanks for mentioning this. Our original paper ignored the compatibility of S1 and S3, but the reviewer is correct that we should add this to the paper. S3 is tighter than S1 for connected graphs (as shown in Lemma 2). The reason we use both S1 and S3 is that S1 can force a connected graph.
>
> **Lemma 2** For any undirected, connected graph $G=(V,E)$, if an indexing satisfies (S3), then it satisfies (S1).
>
> We briefly show their compatibility here and will add Lemma 2 with proof in the final paper. Given any connected graph with an indexing satisfying (S3), we can prove that this indexing satisfies S1 using mathematical induction: node $1$ has to be a neighbor of node $0$, node $2$ has to be a neighbor of node $0$ or $1$, and so on.
>
> **Weakness 2 [Compatibility between constraints]**\
> We appreciate this suggestion. We did not (but should) clarify the interplay between structural constraints and symmetry-breaking constraints. We clarify the compatibility here and will add this explanation in the final paper.
>
> Structural constraints C1 - C25 force graph structures and feasible features that are compatible with molecular design. To present a general formulation, C1 - C25 support controlling the existence of each node by adding variables $A_{v,v},v\in [N]$, which means that we can design molecules with fewer than $N$ atoms. In our analysis and numerical experiments, however, we set $A_{v,v}=1,v\in [N]$ for simplicity. Under such setting, only C5, which corresponds to S1, is related to graph indexing (C1 - C4 and C6 - 25 are irrelevant to indexing). Without assuming $A_{v,v}=1,v\in [N]$, we can add an extra term on C26 ($2^F\cdot (1-A_{v,v})$) to exclude non-existing nodes:
> $$
> \begin{equation*}
>     \begin{aligned}
>         \sum\limits_{f\in [F]}2^{F-f-1}\cdot X_{0,f}\le \sum\limits_{f\in [F]} 2^{F-f-1}\cdot X_{v,f} + 2^F\cdot (1-A_{v,v}),~\forall v\in [N]\backslash\{0\}.
>     \end{aligned}
> \end{equation*}
> $$
> By doing that, we are still considering the indexing for all existing nodes without being influenced by non-existing nodes.
>
> For illustration purposes, we can view CAMD as two separate challenges, where the first one uses structural constraints to design reasonable molecules (including all symmetric solutions), and the second one removes symmetric solutions.

---

> > ### Comment · Reviewer_eveB · 2023-08-16
> >
> > Thank you for your response. Could you please provide a full proof for the proposed Lemma 2?

---

> > > ### Author Response · Authors · 2023-08-16
> > >
> > > Please see the full proof as follows:
> > >
> > > **Proof of Lemma 2** Given any connected graph $G=(V,E)$ with any indexing that satisfies (S3), we prove that this indexing also satisfies (S1) by induction.
> > >
> > > Node $0$ itself is a connected graph. Assume that the induced subgraph consists of nodes $0,1,\dots, v$ is connected, it suffices to show that the induced subgraph consists of nodes $0,1,\dots, v+1$ is connected. Equivalently, we need to prove that there exists $u<v+1$ such that $A_{u,v+1}=1$.
> > >
> > > Assume that $A_{u,v+1}=0,\forall u<v+1$. Since the graph is connected, there exists $v'>v+1$ such that:
> > > $$
> > > \begin{equation*}
> > >     \begin{aligned}
> > >         \exists u<v+1,s.t.A_{u,v'}=1
> > >     \end{aligned}
> > > \end{equation*}
> > > $$
> > > Then we know:
> > > $$
> > > \begin{equation*}
> > >     \begin{aligned}
> > >         \mathcal N(v+1)\cap [v+1]=\emptyset, \mathcal N(v')\cap [v+1]\neq \emptyset
> > >     \end{aligned}
> > > \end{equation*}
> > > $$
> > > Recall the definition of $LO(\cdot)$, we have:
> > > $$
> > > \begin{equation}
> > >     \begin{aligned}
> > >         LO(\mathcal N(v+1)\cap [v+1])>LO(\mathcal N(v')\cap [v+1])
> > >     \end{aligned}
> > > \end{equation}
> > > $$
> > >
> > > Since the indexing satisfies (S3), then we have:
> > > $$
> > > \begin{equation*}
> > >     \begin{aligned}
> > >         LO(\mathcal N(w)\backslash\{w+1\})\le LO(\mathcal N(w+1)\backslash \{w\}),\forall v<w<v'
> > >     \end{aligned}
> > > \end{equation*}
> > > $$
> > > Applying Property 3 gives:
> > > $$
> > > \begin{equation*}
> > >     \begin{aligned}
> > >         LO((\mathcal N(w)\backslash \{w+1\})\cap [v+1])\le LO((\mathcal N(w+1)\backslash \{w\})\cap [v+1]),\forall v<w<v'
> > >     \end{aligned}
> > > \end{equation*}
> > > $$
> > > Note that $w>v$. Therefore,
> > > $$
> > > \begin{equation*}
> > >     \begin{aligned}
> > >         LO(\mathcal N(w)\cap [v+1])\le LO(\mathcal N(w+1)\cap [v+1]),\forall v<w<v'
> > >     \end{aligned}
> > > \end{equation*}
> > > $$
> > > which means that:
> > > $$
> > > \begin{equation}
> > >     \begin{aligned}
> > >         LO(\mathcal N(v+1)\cap [v+1])\le LO(\mathcal N(v')\cap [v+1])
> > >     \end{aligned}
> > > \end{equation}
> > > $$
> > > The contradiction between $LO(\mathcal N(v+1)\cap [v+1])$ and $LO(\mathcal N(v')\cap [v+1])$ completes the proof.

---

> > > > ### Comment · Reviewer_eveB · 2023-08-16
> > > >
> > > > Thank you for your reply!
> > > > Can you elaborate on the derivation of the following step:
> > > > Note that $w>v$. Therefore,
> > > > \begin{equation*}
> > > > L O(\mathcal{N}(w) \cap[v+1]) \leq L O(\mathcal{N}(w+1) \cap[v+1]), \forall v<w<v^{\prime}
> > > > \end{equation*}
> > > >
> > > > Specifically, are you using $L O(\mathcal{N}(w+1) \cap[v+1])= L O((\mathcal{N}(w+1) \backslash w) \cap[v+1])$? If so, it seems the corner case where $w = v+1$ needs additional details.

---

> > > > > ### Author Response · Authors · 2023-08-16
> > > > >
> > > > > Yes. Note that [v+1]={0,1,...,v}. So w and w+1 will never in this set.

---

> > > > > > ### Comment · Reviewer_eveB · 2023-08-16
> > > > > >
> > > > > > I see; thank you for the clarification. I am happy to raise the rating to 7.

---

> > > > > > > ### Author Response · Authors · 2023-08-16
> > > > > > >
> > > > > > > We really appreciate that! Thanks again for your suggestions, which are quite helpful to improve our theory.

---

### Author Rebuttal · Authors · 2023-08-08

To all reviewers,

Thank you very much for taking your precious time to review our paper and give great comments! Your feedback is really helpful for us to clarify and improve our work. More importantly, we are very lucky to have four reviewers focus on different aspects of our paper:

- Reviewer eveB identified the lack of an explanation of the compatibility between constraints (e.g., between S1 and S3, between C1 - C25 and S2 - S3).

- Reviewer ni19 commented on the improvement (e.g., how many symmetric solutions are removed, diversity of solutions) and generality of our proposed approach.

- Reviewer 5u3v asks about the relationship between our work and the symmetry-breaking literature (e.g., how does the literature detect and break symmetry, what are the connections and differences between our techniques and the literature).

- Reviewer eLVB questioned  the experiments (e.g., experimental verification of molecular constraints, baseline, complexity analysis) and the originality of the big-M formulation.

As stated in each separate response, we will add the following clarifications to improve the paper based on the feedback of reviewers:

- A lemma after Theorem 1 will show the compatibility of S1 and S3 (as shown in the response to Reviewer eveB). A paragraph at the end of Section 3.1 will clarify that C1 - C25 will not affect the graph indexing.

- More information about the experiments in Section 3.2, including the purpose, the baseline, the percentage of removed symmetric solution, and the explanation of time out (as shown in the response to Reviewer ni19 and eLVB).

- A literature review about symmetry-breaking in Section 1, including the references we use in the response to Reviewer 5u3v. More discussion about the connections and differences between our approach and the literature in Section 2.4.

- Details about the generality of big-M formulation in Section 2.5.2, and the proposed symmetry-breaking constraints in Section 2.4 and Section 4 (as shown in the response to Reviewer ni19 and eLVB).

Thanks again for your comments, which are more carefully addressed in each separate response.

---

### Decision · Program_Chairs · 2023-09-21

**Decision:**

Accept (poster)

**Comment:**

The paper addresses the issue of GNN optimization and demonstrates practical utility through molecular design application. The authors provide convincing experimental results supporting their claims, underscoring the proposed method's effectiveness. The manuscript is well-written, presenting complex ideas with notable clarity.